# Does Nonstationarity in Rainfall Require Nonstationary Intensity-Duration-Frequency Curves?

Poulomi Ganguli[1,2], Paulin Coulibaly[1]

[1]Department of Civil Engineering, McMaster Water Resources and Hydrologic Modelling Group, McMaster University, 1280 Main Street West, Hamilton, ON L8S 4L7, Canada
[2]Present Address: GFZ German Research Centre for Geosciences, Section 5.4 Hydrology, 14473 Potsdam, Germany

*Correspondence to*: Poulomi Ganguli (poulomi.ganguli@alumnimail.iitkgp.ac.in; gangulip@mcmaster.ca)

**Abstract.** In Canada, risk of flooding due to heavy rainfall has risen in recent decades; most notable recent examples include July 2013 storm in Greater Toronto region and May 2017 flood of Toronto Island. We investigate nonstationarity and trends in the short-duration precipitation extremes in selected urbanized locations in Southern Ontario, Canada, and evaluate the potential of nonstationary Intensity-Duration-Frequency (IDF) curves, which form an input to civil infrastructural design. Despite apparent signals of nonstationarity in precipitation extremes in all locations, the stationary versus nonstationary models do not exhibit any significant differences in the design storm intensity, especially for short recurrence intervals (up to 10-year). The signatures of nonstationarity in rainfall extremes do not necessarily imply the use of nonstationary IDFs for design considerations. When comparing the proposed IDFs with current design standards, for return periods (10-year or less) typical for urban drainage design, current design standards require an update up to 7%, whereas for longer recurrence intervals (50 - 100-year), ideal for critical civil infrastructural design, updates ranging between ~ 2 to 44% are suggested. We further emphasize that above findings need re-evaluation in light of climate change projections since intensity and frequency of extreme precipitation are expected to intensify due to global warming.

## 1 Introduction

Short-duration extreme rainfall events can have devastating consequences, damage to crops and infrastructures, leading to severe societal and economic losses in Canada (CCF, 2013; TRCA, 2013). In a warming climate, extreme precipitation events are expected to intensify due to moistening of the atmosphere (Donat et al., 2016; Fischer and Knutti, 2016; Pendergrass et al., 2015; Prein et al., 2016; Pfahl et al., 2017). Using observational record, review of the literature suggests a dependency between mean and extreme precipitation on temperature (O'Gorman, 2015). The increased water-holding capacity of warmer air, as governed by the Clausius-Clapeyron (C-C) relation

(Lenderink and van Meijgaard, 2008; O'Gorman and Schneider, 2009; Wasko and Sharma, 2015, 2017), intensifies heavy rainfall at a rate of approximately 7-8% $°C^{-1}$ of warming. On a local scale, for sub-hourly and up to six-hourly extreme precipitation, increases at or above the C-C rate have been found in the Netherlands (Lenderink and van Meijgaard, 2008; Lenderink et al., 2017), Switzerland (Ban et al., 2014), Germany (Berg et al., 2013), the UK (Blenkinsop et al., 2015), the Mediterranean (Drobinski et al., 2016), most of Australia (Wasko and Sharma, 2015, 2017; Schroeer and Kirchengast, 2017), North America (Shaw et al., 2011) and China (Miao et al., 2016), while in India (Ali and Mishra, 2017) and northern Australia (Hardwick Jones et al., 2010) negative rates have been reported. The extent of urbanization also contributes to extreme regional precipitation through urban heat island effect and aerosol concentration (Dixon and Mote, 2003; Mölders and Olson, 2004; Mohsen and Gough, 2012; Wang et al., 2015). One of the first attempt to derive nonstationary IDF through Bayesian Inference (BI) approach for extreme value analysis was by Cheng and AghaKouchak (2014), where authors introduced a linear trend in the parameters of the selected distribution. Agilan and Umamahesh (2017) used six physical processes, namely, time, urbanization, local temperature changes, annual global temperature anomaly (as an indicator of global warming), El Niño-Southern Oscillation (ENSO) and Indian Ocean Dipole (IOD) as covariates for the nonstationary extreme precipitation analysis in the city of Hyderabad, India. Their analysis indicated that the local processes, urbanization and local temperature changes are the best covariates for short-duration rainfall, whereas global processes, such as global warming, ENSO cycle and IOD are the best covariates for the long duration rainfall. In their study, time was never qualified as the best covariate for modeling local scale extreme rainfall intensity. Singh et al. (2016) performed nonstationary frequency analysis of Indian Summer Monsoon Rainfall extreme (ISMR; defined as cumulative rainfall over continental India during 1 June to 30 September) and found evidence of significant nonstationarity in ISMR extremes in urbanizing or developing-urban areas (transitioning from rural to urban), as compared to completely urbanized or rural areas. However, their analysis was performed at a spatial resolution of 1° using gridded daily precipitation data obtained from Indian Meteorological Department (IMD). Ali and Mishra (2017) showed that a strong (higher than C-C rate) positive relationship exists between short-duration rainfall extremes, dew point and tropospheric temperature (T850; or the temperature in the upper troposphere at 850 hPa) over 23 urban locations in India. The latter two were subsequently used as covariates for nonstationary design storm estimates. The results indicated an increase in rainfall maxima at a majority of locations assuming nonstationary conditions over stationary atmospheric conditions. In contrast, in another studies, over Melbourne and Victoria, in Australia, Yilmaz et al. (2014; 2017) found superiority of stationary models over nonstationary models. For developing nonstationary models, authors (Yilmaz et al. 2014; 2017), considered both the time dependency and

dependency to large scale climate oscillations affecting Australian rainfall. Using temperature as a covariate for nonstationarity, Wasko and Sharma (2017) investigated the sensitivity of extreme daily precipitation and streamflow to changes in daily temperature. Their results suggested a little evidence of an increase in streamflow with an increase in heavy rainfall events at higher temperature.

However, most of these previous studies have analyzed changes in expected point estimates of nonstationary versus stationary Design Storm Intensity (hereafter referred as DSI), but have not reported the statistical significance of the difference between the two methods of estimates. To our best knowledge, no thorough comparison of stationary vs. nonstationary methods for deriving IDF statistics has been conducted in Southern Ontario, Canada. For densely populated Southern Ontario, observations and multiple climate models suggest increasing trends in regional surface

temperature and extreme precipitation in recent decades (Stone et al., 2000; Paixao et al., 2011; Mailhot et al., 2012; De Carolis, 2012; Burn and Taleghani, 2013; Shephard et al., 2014; Deng et al., 2016). A recent study shows an increase in local surface temperature of $3.06 \pm 0.18$ °C/century in Greater Toronto Area (GTA) since the 1960s (Berkeley Earth, 2017). In July 2013, a single storm event has resulted in 126 mm of rainfall in GTA causing total insured losses of around $940 million and claimed to be the third-most expensive weather-related event in Canada

(CDD, 2015; TRCA, 2013).

Extreme rainfall statistics are often mathematically expressed using the concept of exceedance probability or $T$-year return period [*i.e., $T = 1 / (1 - Fp(P))$*, where $Fp(P)$ is the cumulative probability of the underlying distribution], and graphically as a decision relevant metrics in the form of Intensity-Duration-Frequency (IDF) curves (or relations) (ASCE, 2006; CSA, 2010; EC, 2012). These curves are based on a comprehensive statistical

analysis of historical rainfall records and widely used for the design and operation of storm-water and sewerage systems, and other engineered hydraulic structures (Coulibaly and Shi, 2005; Durrans and Brown, 2001; Lima et al., 2016; Madsen et al., 2009; Rana et al., 2013; Sandink et al., 2016; Yilmaz et al., 2014a). At a given return period and the storm duration, the average DSI is determined from the IDF relationship. The IDF curves are based on fitting a theoretical probability distribution to short-duration (sub-hourly, hourly and daily) Annual Maximum

Precipitation (AMP). The approach can be implemented both locally (at site) or regionally [Svensson and Jones, 2010; Regional Frequency Analysis (RFA) or pooled]. The RFA is used when available record lengths are short or at locations where no observed data are available (Castellarin et al., 2012; Komi et al., 2016). However, various RFA estimation methods have certain drawbacks; for instance, the index flood method is sensitive to the homogeneity assumption and formation of regions; in a Bayesian method of regionalization, the prior distributions

of parameters are often not precise enough and do not add precision to the estimates. Komi et al. (2016) summarize limitations and advantages of some of the widely used RFA techniques. In the present study, the available records across all sites range between 47 and 66 years, which are more than the climatology (often over time periods of 30-years) of a region. Therefore, we employ at-site frequency analysis herein. This also allows a consistent comparison

with the Environment Canada (EC) IDFs that have been used in practice in the study area. For Canada, information for preparation of IDFs and nation-wide IDF curves are archived at EC Engineering database (Environment and Climate Change Canada, 2012; http://climate.weather.gc.ca/prods_servs/engineering_e.html), which are produced based on short-duration available rainfall records from the Tipping-Bucket Rain Gauges (TBRG). Nevertheless, the methodology to derive existing IDF curves has certain drawbacks, such as, the current IDF curves in Canada

are based on the assumption of stationarity, which implies statistical properties of hydroclimatic time series will remain same over the period of time.  However, the impact of urbanization and human-induced climate change (Field, 2012; Milly et al., 2008; Villarini et al., 2009) raise the question whether the stationarity assumption to derive IDF curves is still reliable for urban infrastructural planning (Sarhadi and Soulis, 2017; Cheng and AghaKouchak, 2014; Jakob, 2013; Yilmaz et al., 2014a; Yilmaz and Perera, 2013).

The nonstationary behavior of rainfall extremes is already being reflected in the increase in frequency or magnitude of such events, resulting in a shift of its distribution [Figure SPM 0.3 in Intergovernmental Panel on Climate Change Special Report on Extremes, IPCC SREX Report: Field, 2012; Fig S1: IPCC AR5 working Group Report, (Stocker et al., 2013)]. For instance, seasonal and annual extreme precipitation in north-central and eastern US in 2013 (Knutson et al., 2014); extreme rainfall in the Golden Bay region in New Zeeland (Dean et al., 2013); increase in

summer precipitation rate in northern Europe (Yiou and Cattiaux, 2013); successive winter storm events in southern England in 2013/2014 leading to severe winter floods (Schaller et al., 2016), are primarily attributable to intrinsic natural variability and partly to anthropogenic influences. The asymmetric changes in the distribution of extremes owing to climate change have been subsequently validated for winter temperature extremes over the northern hemisphere (Kodra and Ganguly, 2014), and regional short-duration precipitation extremes in India and Australia

(Mondal and Mujumdar, 2015; Westra and Sisson, 2011). Two of the recent studies (Deng et al., 2016; Mailhot et al., 2012) analyzed large ensemble of CMIP3 Global Climate Model (GCM) runs and a sub-set of regional climate models that are part of North American Regional Climate Change Assessment Program (NARCCAP) in terms of impact-relevant metrics over Canada. Both studies confirmed a relative increase in intensity and magnitude of rainfall extremes, especially over Southern Ontario. This issue has come to attention in the Guidelines for Canadian

Water Resources Practitioner (CSA, 2010), that urges the need for updated IDF calculations: "…*climate change will likely result in an increase in the intensity and frequency of extreme precipitation events in most regions in the future. As a result, IDF values will optimally need to be updated more frequently than in the past* ….".

Furthermore*,* so far very few studies have reported the difference between the updated versus EC generated IDFs, taking into account nonstationarity in design consideration. Simonovic and Peck (2009) compared updated versus EC IDFs for the city of London, Ontario and reported EC IDF curves shows a difference of the order of around 20%. However, their analysis was based on the stationarity assumption of precipitation extremes. Similarly, Coulibaly et al. (2015) have compared EC IDFs with stationary GEV based IDF curves across Southern Ontario, however, no nonstationary methods were investigated. Motivated with these research gaps, here we address several important questions pertained to short-duration precipitation extremes over Southern Ontario, to improve pro-active management of storm-induced urban flooding. First, is there any signature of statistically significant nonstationary trends (gradual or monotonic changes), change points or regime shifts (occurrence of any abrupt changes in mean/variance of the distribution) in short-duration AMP in densely and moderately populated urbanized locations across Southern Ontario? Second, does nonstationarity in the time series necessitates the use of nonstationary IDFs, barring economic consideration and mathematical complexity involved in the design? Third, how can we use this knowledge to assess the credibility of existing EC-generated IDFs in the backdrop of a changing climate? We do not attempt to provide a methodological comparison of EC-generated versus current approach but will focus on differences in estimated DSI values between the updated and EC-IDF. Further, to this end, we test the hypothesis that signatures of nonstationarity in rainfall extremes do not necessitate the use of nonstationary IDFs for design considerations. In general, urban drainage areas have substantial proportions of impervious or semi-impervious land cover, which significantly reduce response time to extreme precipitation and increase the peak flow, resulting into storm-induced floods (Miller et al., 2014). Hence, it is the short-duration precipitation extremes, which controls the design of urban infrastructure (Mishra et al., 2012). Therefore, we focus our analysis on AMP intensity. We select Southern Ontario as a test bed because of the majority of stations with more than 30-years of available rainfall record (Adamowski and Bougadis, 2003; Deng et al., 2016; Shephard et al., 2014). Recent studies have indicated that the region is more vulnerable to climate change than any other part of Canada (Deng et al., 2016; Mailhot et al., 2012). Furthermore, southern Ontario is one of the prominent economic hubs with largest population concentration in Canada (Bourne and Simmons, 2003; Kerr, 1965; Partridge et al., 2007). In this context, we

explore a robust statistical framework to evaluate possible nonstationary trends, analyze the frequency of urban precipitation extremes and assess the risk of severe rain-induced urban flooding in Southern Ontario (Table 1).

## 2 Study Area and Data

### 2.1 Study Area

Southern Ontario is situated on a Southwest-northeast transect, in the southernmost Canadian region, and separated from the United States by lakes Erie, Huron, and Ontario (Figure 1). The study area includes nine densely and moderately populated urbanized and anthropogenically altered locations of the Windsor - Kingston corridor (Figure 1; Table 1). The last column in Table 1 shows a list of missing years and AMP values for each duration at each station. The Digital Elevation Model (DEM) of the study area was derived from Shuttle Radar Topography Mission

(SRTM) 90-m Digital Elevation Database v4.1 (Jarvis et al., 2008), which indicates a shallow slope with a maximum altitude of 670 m above Mean Sea Level (MSL). The proximity to Great Lakes and topographic effect, especially in areas to the lee of Lakes Erie, Lake Ontario, and the Georgian Bay significantly modifies the climate in the region (Baldwin et al., 2011). Convective showers and thunderstorms primarily modulate the summer rainfall, but fall rainfall is dominated by reduced convective activity and increased lake effect precipitation (Lapen

and Hayhoe, 2003). Further, the topographic features and associated westerly winds in the Niagara Escarpment and the Oak Ridge Moraine, play a significant role in modulating rainfall in Toronto region. On the other hand, Windsor metropolitan area, the southernmost urbanized location in the region, has a humid continental climate, which results in warm summer temperature (30°C or higher) with the greatest precipitation in the spring and summer seasons, and lowest in the fall and winter (Sanderson and Gorski, 1978). Moreover, because of the part of Windsor-Detroit

international transborder agglomeration, the extreme summer precipitation in the city of Windsor is primarily influenced by convection and urban heat island effect (Sanderson and Gorski, 1978; De Carolis, 2012).

### 2.2 Hydrometeorological Data

We identified the station locations (Figure 1b) based on the quality of long-range rainfall records (*e.g.*, 30 years or

25 more) and 2011 Census information archived at Statistics Canada website (https://www12.statcan.gc.ca). The geographic areas of these locations are extracted from 2011 census digital boundary shape files (https://www12.statcan.gc.ca/census-recensement/2011/geo/bound-limit/bound-limit-2011-eng.cfm). The Toronto metropolitan area is the most populous (over 5 million population) and known to be one of the fastest growing

population base in Canada (http://torontosvitalsigns.ca/main-sections/demographics/), while Fergus is the least populated (population of around 19,000) (Table 1) city. The other cities have population ranges between ~ 500,000 (Hamilton) and 30,000 (Stratford) (Table 1). We obtained AMP observations at particular durations (15-, 30-minutes, 1-, 2-, 6-, 12- and 24-hours) with a few data gaps from Canada's National Climate Data Archive,

maintained by the EC (http://climate.weather.gc.ca/prods_servs/documentation_index_e.html). The rainfall records collected from TBRG are thoroughly quality controlled (Shephard et al., 2014). These records have been previously analyzed for the assessment of national extreme rainfall trends (Burn and Taleghani, 2013; Shephard et al., 2014). We consider seven storm durations ranging from 15-, 30- minutes (the typical time of concentration for small urban catchments), and 1-, 2-, 6-, 12-, and 24- hours (the standard time of concentration for larger watersheds)

following a previous study (Bougadis and Adamowski, 2006). Except for a few stations (for example, Toronto International Airport and Trenton Airport), for most of the sites, the AMP observation is available either until the year 2007 or before (Table 1). Also, we found missing values in the AMP time series in all sites. We obtained daily and hourly rainfall records from the EC website and Toronto Region Conservation Authority (TRCA).

## 3 Methods

Figure 2 shows schematics of the overall analysis. In subsequent subsection, we will discuss each of these steps in detail.

### 3.1 Infilling Missing AMP Record

We infilled missing values and updated the AMP records by successively disaggregating daily rainfall values to hourly and sub-hourly time steps using Multiplicative Random Cascade (MRC)-based disaggregation tool. The

Cascade-based disaggregation model for continuous rainfall time series was suggested by (Olsson, 1995, 1998). The technique was later successfully implemented by (Güntner et al., 2001; Jebari et al., 2012; Rana et al., 2013) for temporal disaggregation of point rainfall and the development of IDF-curves from short-duration rainfall extremes. Due to freezing weather conditions during winter, most of the TBRGs' are inoperative from early November to late April of the following year. Therefore, when short-duration rainfall records were not available,

the AMP values over moving windows of $n$- durations ($n$ varies from 15-, 30- minutes and 1-,2-,6-,12- and 24-hours) are extracted from May to October (warm season) disaggregated rainfall volumes for remaining years. There are several reasons for selecting warm periods: first, extreme rainfall events mostly occur in the study area during the warm season (Cheng et al., 2010); second, the focus of our analysis is an investigation of extreme rainfall

related flood risks and development of IDF curves using extreme rainfall statistics. We adjusted the occasional overestimation of extreme values at a higher order cascade step by a statistical post-processing method. We employed Quantile Matching (QM) approach (Li et al., 2010), which claims to outperform other simple bias correction methods and corrects not only the mean but also the variance of the distribution of interest

(Gudmundsson et al., 2012; Teutschbein and Seibert, 2012). QM is based on equidistant cumulative probability distribution matching of observed and disaggregated AMP time series using three-parameter Generalized Extreme Value (GEV) distribution. Although like other statistical post-processing technique, QM relies on the stationarity assumption of the time series, in our case, we applied QM to entire time series of both observed and disaggregated AMP, which comes from the same station location (or similar spatial resolution) and a similar period. Therefore,

we avoid potential consequences of inflation by quantile mapping (Maraun, 2013) in our analysis. We discuss the implementations of MRC, adjustment of extremes and associated model fits in more details in the Supplementary Information (SI 1).

## 3.2 Detection of Nonstationarity

A series of statistical tests are employed to detect the presence of nonstationary trends and abrupt shifts in the short-

15 duration AMP before frequency analysis. The multiple tests allow a more rigorous and comprehensive assessment of overall trend in the time series since certain tests are complementary to each other (Sadri et al., 2016; Yilmaz et al., 2014, 2017). Figure 2 shows schematics of the overall analysis. Most of the trend and change-point detection algorithms assume observations are mutually independent. The presence of autocorrelations in the time series, over (or under) -estimates the statistical significance of trend and change-point detection algorithms (Serinaldi and

20 Kilsby, 2016; von Storch and Navarra, 1999). We employed a Ljung-Box test with 20 lags to the short-duration AMP time series at each site to check if they show statistically significant autocorrelation (at 5% and 10% significance levels). For the time series with no serial autocorrelation, we test for trending behavior and nonstationarity. It is also important to note that presence of nonstationarity may not be evaluated merely on the basis of trends or abrupt shifts in the time series, even if the increasing or decreasing trends are statistically

significant (Yilmaz et al., 2014). First, we check for a presence of nonstationarity in the time series by employing a unit root-based Augmented Dickey-Fuller (ADF; Dickey and Fuller, 1981) test. However, the test may have a low power against stationary near unit root processes (Dritsakis, 2004; Chowdhury and Mavrotas, 2006). Therefore, as a complementary to unit root test, KPSS test (Kwiatkowski et al., 1992) is employed to validate the results of the ADF test. Since both ADF and KPSS tests assume linear regression or normality of the distribution;

alternatively, a log-transformation can convert a possible exponential trend present in the data into a linear trend. Therefore, following previous studies (Gimeno et al., 1999; Van Gelder et al., 2006), AMP time series is log-transformed before applying stationarity tests. However, Yilmaz et al. (2014) did not observe the presence of any significant nonstationarity in extreme rainfall time series in the city of Melbourne even after employing ADF and

KPSS tests. Therefore as an alternative, we also employed frequency-based Priestley and Subbarao test ['PSR'-test; (Priestley and Rao, 1969), which is able to better capture nonlinear dynamical nature of hydrological system than that of the former two tests (Ali and Mishra, 2017; Hamed and Rao, 1999). Next, we detected the presence of smooth and abrupt changes in the time series. The continuous or monotonic trends in short-duration rainfall extremes are identified using non-parametric Mann-Kendall trend statistics with correction for ties (Hamed and

Rao, 1998; Reddy and Ganguli, 2013) at 5 and 10% significance levels. In general, the abrupt change (or change point) in the time series occurs at a single point in the record and bifurcates the time series into two halves, either with different means, variances, or both dissimilar means and variance together at each part. The change-point in location (or mean) is identified using non-parametric Pettit's (Pettitt, 1979) and Mann-Whitney tests (Ross et al., 2011). As indicated by previous studies (Xie et al., 2014; Yue and Wang, 2002), the rank-based nonparametric

Mann-Whitney test is not really distribution free and the power of the test is often affected by the properties of sampled data. In practice, when a real change point is unknown, often Mann-Whitney test, in general, does not work well, and the Pettitt method can yield plausible change point location along with its statistical significance. However, the significance of the Pettitt test can be obtained using an approximated limiting distribution (Xie et al., 2014; SI2).The shift in scale (or variance) is detected using non-parametric Mood's Test (Ross et al., 2011; See

Figure 2 for details). We applied nonparametric tests due to their robustness to non-normality, which usually appears in the hydroclimatic time series. Further, in order to reduce the number of underlying assumptions required for testing a hypothesis, such as the presence of a specific kind of trend or change point in the data, nonparametric tests are employed. For the time series with significant autocorrelation, we employed a Trend-Free Pre-Whitening procedure (TPFW; SI 2) as described in (Yue et al., 2002, 2003) and later modified by (Petrow and Merz, 2009).

Then, we applied trend and change point detection algorithms to the pre-whitened AMP extremes. In order to test the issue of multiple comparisons associated with statistical analysis, we analyzed p-values of five statistical tests, i.e., Ljung-Box test, KPSS test, Mann-Kendall trend test, Priestley-Subbarao test, and Pettitt test using False Discovery Rate (FDR) method (not shown here) as suggested by (Benjamini and Hochberg, 1995). However, we excluded ADF, Mann-Whitney and Mood tests from the analysis since unlike other tests, the higher p-value in

ADF statistics indicate the presence of nonstationarity in the time series. On the other hand, the latter two tests do not offer any p-values.

**3.3 Extreme Value Analysis of Sub-daily and Daily Precipitation Extremes**

Nation-wide EC IDF curves were developed using a particular family of distribution function from the extreme value theory (*i.e.*, Gumbel distribution or Extreme Value type I, hereafter referred as EVI). However, EV1 distribution has certain limitations, such that it is a non-heavy tailed distribution and characterized by constant skewness and kurtosis coefficients (Markose and Alentorn, 2005; Pinheiro and Ferrari, 2016). However, the short-duration AMP intensities often exhibit fat-tail behavior, indicating large skewness and kurtosis. In fact, a few

studies in the past have shown that EV1 fits poorly to the historical rainfall extremes (Burn and Taleghani, 2013; Coulibaly et al., 2015). Therefore, in the present study, we perform frequency analysis of extreme precipitation using GEV distribution. The choice of GEV distribution was based on previous studies where various distribution functions were compared in the study area (Coulibaly et al. 2015; Switzman et al., 2017). GEV distribution is a combination of Gumbel, Fréchet and Weibull distributions and is fitted to block or AM time series (Cheng and

AghaKouchak, 2014; Katz et al., 2002; Katz and Brown, 1992). The GEV distribution is characterized by three parameters, the location, the scale and the shape of the distribution, which describes the center of the distribution, the deviation around the mean and the shape or the tail of the distribution (Katz et al., 2002; Katz and Brown, 1992). The cumulative distribution function of stationary (time-invariant) GEV model is given by (Coles et al., 2001; Gilleland and Katz 2016):

$$G(z) = \begin{cases} \exp\left\{ -\left[ 1 + \zeta\left( \frac{z - \mu}{\sigma} \right) \right]_+^{-1/\zeta} \right\} & if \ \zeta \neq 0 \\ \exp\left\{ -\exp\left( -\frac{z - \mu}{\sigma} \right)_+ \right\} & if \ \zeta \to 0 \end{cases} \tag{3.1}$$

Where, $y_+ = \max\{y, 0\}$, and

$z \in \left[(\mu - \sigma)/\zeta, +\infty\right)$ when $\zeta > 0$; $z \in \left(-\infty, (\mu - \sigma)/\zeta\right]$ when $\zeta < 0$; and $z \in (-\infty, +\infty)$ when $\zeta = 0$

$\mu$ is a location parameter, $\sigma$ is a scale parameter and $\zeta$ is a shape parameter determining the heaviness of the tail. The shape parameter $\zeta$, determines the higher moments of the density function and also the skew in the

probability mass. The '+' sign indicates positive part of the argument.  The Eq. (3.1) encompasses three types of DFs based on the sign of the shape parameter, $\zeta$ : (i) the Fréchet, with a finite lower bound of $(\mu - \sigma)/\zeta$ and an unbounded, heavy upper tail, ($\zeta > 0$), (ii) the Weibull, unbounded below and with a finite upper bound of $(\mu - \sigma)/\zeta$, ($\zeta < 0$) and (iii) the Gumbel, unbounded below and above with a light upper tail $\zeta = 0$, formally obtained by taking limit as $\zeta \to 0$. The Gumbel distribution is described by an unbounded light tailed distribution

and the tail decreases rapidly following an exponential decay. The Fréchet distribution is a heavy-tailed distribution, and the tail drops relatively slowly following a polynomial decay (Towler et al., 2010). On the other hand, the Weibull distribution is a bounded distribution. Here we compare the performance of both stationary and nonstationary form of GEV distribution. For stationary model, we estimate  parameters using BI coupled with Differential Evaluation Markov Chain (DE-MC) Monte Carlo (MC) simulation as suggested by (Cheng et al., 2014;

Cheng and AghaKouchak, 2014). For nonstationary model, the shape parameter is assumed as constant throughout. Here it should be noted that for modeling temporal changes in $\zeta$ requires long-term observations, which are often not available in practice (Cheng et al., 2014). Hence, following previous studies (Cannon, 2010; Cheng et al., 2014; El Adlouni et al., 2007; Gu et al., 2017) we incorporate time- varying covariates into GEV location (GEV$_t$-I), and both in location and scale parameters (GEV$_t$-II) respectively, to describe trends as a linear function of time (in

20   years):

$$\mu(t) = \mu_1 t + \mu_0 \tag{3.2}$$

$$\sigma(t) = \sigma_1 t + \sigma_0 \tag{3.3}$$

Since the scale parameter must be positive throughout, it is often modelled using a log link function (Gilleland and Katz, 2014)

$$\ln \sigma(t) = \sigma_1 t + \sigma_0 \;\Rightarrow\; \sigma(t) = \exp(\sigma_1 t + \sigma_0) \tag{3.4}$$

Where $t$ is the time (in years), $\lambda = \left\{ \mu_1, \mu_0, \sigma_1, \sigma_0, \zeta \right\}$ are the parameters.

Then we estimate parameters of the nonstationary GEV distribution by integrating BI combined with DE-MC simulation. For AMP intensity, we derive the time variant parameter(s) from the 50[th] (the median or the mid-point of the distribution) percentiles of the DE-MC sampled parameter(s). We obtain the associated 95% credible intervals (the bounds) from the 2.5[th] and 97.5[th] percentiles of the simulated posterior samples (See SI 3 for details). We perform the calculations following (Cheng and AghaKouchak, 2014) using an MATLAB-based software package, Nonstationary Extreme Value Analysis (NEVA, Version 2.0). The Bayes factor followed by Akaike information criterion (*AIC*) with a small sample correction (*AIC$_c$*) are used to identify the best model. The *AICc* claims to avoid overfitting the data as compared to traditional AIC (Burnham and Anderson, 2004; Hurvich and Tsai, 1995). Here we assess model fitness based on a least square sense of AIC statistics considering maximum deviation between empirical (obtained from rank-based plotting position formula) and modelled cumulative distribution (CDF; Dawson et al., 2007; Hu, 2007; Karmakar and Simonovic, 2007, 2007). For calculation of *AICc* statistics, we consider median of the DE-MC sampled parameters, which can be considered as an average or expected value of risk in the historical observation. Besides this, we also assess the performance of models using Probability-Probability (*PP*) plots. The derived model parameters are then utilized to obtain DSI using the concept of a *T*-year return period. We discuss the methods to estimate DSI and *T*-year return periods using stationary and nonstationary methods in detail in section SI 4. To test (statistically) significant difference in the estimated DSI from the best-selected stationary versus nonstationary models, we calculate standardized z-statistics for selected return periods (Madsen et al., 2009; Mikkelsen et al., 2005). We applied the two-sided option with 10% significance levels to assess the statistical significance of the test statistics (See SI 5 for details). Finally, we compared the DSI obtained from nonstationary and stationary models with existing EC-generated DSI estimates.

## 4 Results

The extreme rainfall statistics show high standard deviation with positive skew behaviour. The skewness is a measure of the asymmetry in the AMP distribution. Positive values of skewness indicate that data are skewed to the right. The skewness of sub-hourly precipitation extremes varies between 0.22 and 4.45, with highest being 30-min AMP record at Hamilton and least being at Oshawa respectively. Likewise, for hourly extremes, the skewness ranges between 0.54 and 2.54, with least being 1-hour AMP at Oshawa and highest is 1-hour AMP at Hamilton respectively. The majority of stations show positive excess kurtosis, which indicates the data have a distinct peak

near the mean, which decline rapidly, and have heavy tails. We find the presence of statistically significant autocorrelations in the AMP time series of Toronto International Airport, Hamilton Airport, and Fergus Shand Dam (SI 2). We apply nonparametric TPFW to precipitation extremes with a significant autocorrelation (Tables S4.1, S5.1, and S12). However, two successive TPFWs fail to correct the effect of autocorrelation in 12- and 24-hour

duration extremes in Shand Dam. Hence we exclude those two time series from frequency analysis (Table S12). The ADF-test for nonstationarity is statistically significant in all durations, as indicated by the higher *p*-values. As a complementary to ADF test, we also employed KPSS and PSR tests (Figure 2; SI 2) to check significant nonstationarity. Figure 3 shows the spatial distribution of trends, change points and nonstationarity in short-duration rainfall extremes. We find co-occurrences of trends, change points and nonstationarities in extremes at

multiple locations. In general, the three sites in the extreme Northeast, the Oshawa WPCP, Trenton Airport and Kingston P. Station show evidence of statistically significant upshifts and nonstationarities in the time series, whereas the rest of the sites in the Southwest exhibit downshifts and statistically significant nonstationarities (Figure 3). For 2-hour and beyond durations, London International Airport shows the presence of statistically significant downward trends with change points. An increase (decrease) in mean precipitation imply an increase

(decrease) in heavy precipitation and vice-versa. Further, it could also alter the shape of the right-hand tail, changing overall asymmetry in the distribution (Fig. S1), and hence affecting the nature of extremes (Stocker et al., 2013). Furthermore, the presence of opposite signs of trends within a proximity of sites are prominent in all durations, for example, except for 1-hour duration, extremes in all durations at Toronto International Airport and Oshawa WPCP show downwards and upward shifts respectively. Our findings confirm the other study (Burn and Taleghani, 2013),

where authors report a lack of spatial structures and presence of different trends within a close vicinity of stations. Further, we find statistically significant monotonic increase and abrupt step changes, both in mean and variance in Oshawa and Trenton respectively (Tables S6 and S10), whereas London shows (significant) decrease (Table S9) from 6-hour duration and beyond. A few sites, such as Windsor, Kingston and Stratford show (significant) step changes as confirmed by Mann-Whitney and Mood Tests (Tables S7, S8 and S11) respectively. On the other hand,

Toronto, Hamilton and Fergus Shand Dam (Tables S4, 4.1; S5, 5.1; S12) do not exhibit any significant gradual or abrupt changes in the AMP time series. The ADF tests show the presence of nonstationarity in all durations across the sites. To further validate results of ADF test, KPSS and PSR tests are employed. The KPSS test detects the presence of nonstationarity at 3 out of 9 sites for 24-hour rainfall extreme at 5% significance level, whereas the results of PSR test indicate nonstationarity across 5 sites in 24-hour rainfall extremes. While KPSS test alone could

not detect the presence of nonstationarity in any of the extreme series in Oshawa and Stratford respectively, the

results of PSR test could not discern nonstationarity in any of the short-duration rainfall extreme in Windsor. Both of these tests taken together detect the presence of nonstationarity in rainfall extremes across 6 out of 9 sites. We find even if trends in individual sites may not deem significant, the magnitude of trends (as measured by slope per decade, Tables S4 – S12 in SI2) is never zero in any of the sites. Although we present a range of statistical tests to investigate plausible shifts in the time series we do acknowledge that statistical inferences may be affected by the problem of multiple comparisons resulting into a set of false positive outcomes. On analysing FDR-based multiple comparison procedure, we find that except Windsor, in all sites the presence of trend and non-stationarities in the time series turns out to be significant with the highest number of statistical tests showing adjusted p-value < 0.10 in Trenton followed by Hamilton. This indicates our analysis is not affected by the issue of multiple comparisons.

A weak trend can also have a significant impact on the results of probability analysis (Porporato and Ridolfi, 1998). Hence even if precipitation extremes often exhibit statistically insignificant trends in few durations, we assess the performance of both nonstationary and stationary models in all sites. Tables 2 – 5 list performance of nonstationary versus stationary models for selected airport locations, whereas Tables S13 - S17 present results of the distribution fit for the remaining stations. Barring a few exceptions, the shape parameters in most of the models range between -0.30 and +0.3, which is an acceptable range of GEV shape parameter as shown in an earlier study (Martins et al., 2000). Our results corroborate well with recent research (Papalexiou et al., 2013; Wilson and Toumi, 2005), which showed that distribution with fat tails (with GEV shape parameter, $\zeta < 0$) fits better for the precipitation extremes. The nonstationary models are selected employing Bayes-factor and minimum *AICc* criterion. For example, for the 6-hour duration storm at Hamilton Airport, the nonstationary GEV$_t$-I (nonstationary model with time-varying GEV location) model performed the best as shown by both test metrics. However, in certain cases, nonstationary models do not pass Bayes-factor test. In such cases, we select the best nonstationary model (*i.e.*, between GEV$_t$-I and GEV$_t$-II) following *AICc* test statistics. Overall values of Bayes factor indicate that there is no strong evidence to favor or reject any of the three models. In general, although we find the stationary model cannot be rejected, it does not imply that there is no change. We may be unable to detect the apparent signal of nonstationarity due to the strong natural variability present in the data. However, it should be noted that the objective of the present analysis is to

compare the design storm obtained from stationary versus the best nonstationary model and not to analyze the best distribution between them.

As a measure of uncertainty, we also report the 95% credible interval of design rainfall quantiles at 100-year return period as a ratio between the upper and the lower bounds, which ranges between the factor of 1.2-to-1 and 3.9-to-1 in all cases. The performance of time-varying GEV models (Figure S9) closely follows the spatial pattern as indicated in the trend map (Figure 3). For example, Trenton Airport, which showed significant upward trends with change points and nonstationarity, is better modeled by the nonstationary GEV distributions for most of the durations. Likewise, we find that in a few cases, GEV II fits best if the time series exhibit (significant) evidence of a nonstationarity as detected by PSR-test statistics, for example, 15-min and 12-hr extremes in London and Toronto International Airport (Tables 4 and 7) respectively. However, in many cases, the performance of nonstationary models are often comparable and even superseded by their stationary counterparts (SI 3). In fact, the scatter of data points in the *PP*-plots (Figures S10 – S12) suggests a close resembles between stationary and nonstationary models across all durations. Figure 4 shows the relation between DSI and durations (ranges between 15-min and 24-hr) for 100-year return periods estimated by stationary versus nonstationary GEV distributions. The interquartile range of the boxplot shows the uncertainty in estimated rainfall quantiles obtained using BI. However, the spread of the boxes simulated by the nonstationary model is found to be relatively narrower as compared to the one simulated by the stationary model for most of the sites (Figure 4), indicating less uncertainty in the estimated quantiles. For shorter return periods, such as 10- and 5-year, the DSI from stationary versus nonstationary models shows more or less subtle differences (Figures S13 – S14) than those for the 100-year.

Figure 5 displays the differences in DSI obtained using the best performing nonstationary model relative to the stationary models using percentage changes and z-statistics for different durations and return periods. While percentage change indicates a magnitude of change, the z-statistics show statistical significance of the relative difference in estimated DSI using the two different methods. The percentage differences at 2- and 10-year return period are small relative to larger return periods. At 100-year return period, a maximum positive difference of around 44% is observed at 12-hour storm duration in Toronto International Airport (Table S18.1). The standardized z-statistics show positive (negative) values indicating an increase (decrease) in DSI values assuming nonstationarity in return period estimates against its stationary counterparts. However, a comparison between *T*-year event estimates from both models indicates statistically indistinguishable differences in rainfall intensity. We find for all return periods and durations, z-statistics ranges between -1 and +1 across all nine sites (SI 5). Nonetheless, extreme

precipitation intensity shows either positive or negative (statistically insignificant) changes in signs. The difference between DSI shows a decrease, at 1- and 2-hour storm duration in Toronto, 6-hour storm duration in London, and 15-min and 12-hour storm duration at Trenton Airport for 50- and 100-year return periods (Figure 5, SI 5). In contrast, Toronto, Windsor and London International Airport show an increase in DSI value at 15-min duration

(Figure 5; SI 5), although the increase is statistically insignificant. Further, we note, except 6- and 12-hour storm durations, the performance statistics show a comparable and in few cases even better performance of the stationary versus nonstationary GEV models across most of the sites (SI 3). At 2- and 10-year return periods, which is typical for most urban drainage planning, the differences are close to zero (Figure 5, Tables S27 and S28 in SI 5) for most of the duration.

Figures 6 and 7 compare the $T$-year event estimates of updated versus EC-generated IDFs for different return periods taking into account both stationary and nonstationary condition. The median and associated lower and upper bounds of the ratio of regional updated versus EC-generated $T$-year event estimates can be interpreted as analogous to most likely, minimum and maximum plausible scenarios. While the positive value of the ratio indicates a required increase in DSI, the negative value indicates a decrease in DSI estimate. Considering

nonstationarity, at $T = 10$-year return period (Figure 6, SI 4), the ratio of updated versus old estimates of DSI is in the order of ~ 1.01 – 1.08, which indicates the required increase are in the order of 1.4 to 7.2%. At $T = 2$-year return period, except Oshawa and Windsor, a majority of sites show a decrease in DSI for most of the storm durations (Figure 6, middle row). In contrast, an increase in the estimated ratio is more pronounced at 50- and 100-year return periods, which are in the order of ~ 1.03 - 1.80 (Figure 7, SI 4). While for Toronto International Airport and

Hamilton Airport, we find no increase in the short-duration rainfall extremes of less than 1- hour and 50-year return period considering nonstationary condition, the increase is more pronounced for longer durations and larger return periods (12 and 24-hour, and 50- and 100-year return period, SI 4). For longer recurrence intervals, while the heat maps of minimum bounds and the most likely scenario show a smaller number of stations and durations to reach a ratio of 1.5 and beyond, the maximum bounds suggest a sharp increase in the ratio across all durations and locations.

Further, for return periods of 50-year and more, the increase in the ratio is more prominent in the upper bound of the stationary models (Figure 7, left two columns) as compared to the nonstationary models. The resulting increase in $T$-year event estimates is because of the relatively wider confidence interval of estimated DSIs in stationary models than that of the nonstationary models (Figure 4; SI 3). In general, for larger return periods, our analysis reveals, the increase in the ratio of updated versus EC-generated rainfall intensity is more prominent in sites with

statistically significant signatures of nonstationarity, upward trends, and change points. For example, the updated DSIs of Oshawa WPCP, Windsor and Trenton Airport shows an increase in the ratio for most of the durations and return periods as compared to the EC-generated DSI values (SI 4). On the other hand, except for the 100-year return period events, the hourly precipitation extremes in London International Airport, in general, show a decrease in the ratio (Tables S23.1 – 23.2) across all return period, which is predominantly due to the presence of significant downward trends with change points in the time series.

Based on the study results and in anticipation of stakeholders' participation in adaptive management, we present updated IDFs for four selected urbanized locations across Southern Ontario (Figure 8). In order to distinguish between stationary and nonstationary method of analysis, we also present updated IDF assuming stationary condition relative to EC IDF in the same plot (in top panel). The comparison of remaining sites is presented in Figure S15. Thus we made the first attempt to compare the results of updated versus EC-generated IDFs considering both nonstationary and stationary conditions, which are the part of contemporary Design Standards and widely used by the stakeholders and practitioners. Overall, the updated IDFs closely follow the pattern of trends analogous to EC-generated IDFs, except for the 100-year return period. The difference is more pronounced considering nonstationary condition, especially at Toronto International Airport (Figure 8), Oshawa WPCP and Stratford WWTP (Figure S15). At longer durations and return periods, stations in metropolitan areas (such as Toronto International Airport, Hamilton Airport, Oshawa WPCP and Windsor Airport) show large differences in DSIs, whereas moderately populated locations such as Kingston P. station and Fergus Shand dam show a relatively small change. Considering, nonstationary condition, the maximum increase in Furgas Shand dam is noted as 18.7% for the 2-hour storm duration and 100-year return period, whereas an increase of around 44.5% is shown for 12-hour storm duration at Toronto International Airport. For $T$ = 10-year or less, we find a decrease in the range of ~ 2 – 40% in the $T$-year event estimates (SI 4). Meanwhile, for $T$ = 10-year return period, we find the increase is in the order of ~1.4 to 7.2% across several stations. Considering nonstationarity, at $T$ = 50-year and more, the required increase ranges between ~ 2.8 – 44.5%. We find the largest increase is for the 12-hour rainfall extreme in Toronto International Airport (~ 32 – 44.5%; Table S18.1), followed by 2-hour extreme at Stratford WWTP (~ 27 – 36%; Table S25.1). However, considering stationarity condition, at $T$ = 50-year and more, the required increase ranges between ~ 1.4 – 26%. It should be noted that above results are based on an average risk approach for extreme value analysis considering median of the sampled parameters in the historical observation and not taking into account the overall risk envelope (*i.e.*, minimum and maximum bounds). In summary, our findings confirm that updates in the

order of ~ 2 – 44% are required based on locations and return periods to mitigate the risk of precipitation induced urban flooding irrespective of the choice of methods used in the IDF estimation (SI 4). The results are consistent with (Simonovic and Peck, 2009), in which authors recommend an average update of about 21%, with a difference, range between ~ 11 – 35% for the updated versus EC-IDF in London Metropolitan area. However, they assumed
stationarity condition to develop at-site IDF. The above results also highlight the need to update existing EC IDFs, which are generated using Gumbel probability distributions and do not fit the data well.

## 4 Discussions and Conclusions

This paper has sought to assess signatures of nonstationarity in densely and moderately populated urbanized locations across Southern Ontario, which is one of the major economic hubs in Canada. We update short-duration
rainfall extremes with latest available ground-based observations and present a comprehensive analysis to evaluate nonstationary versus the stationary method of IDF estimation. This analysis yields two principal findings. First, detectable non-stationarity in rainfall extremes does not necessarily lead to significant differences in design storm values. Second, comparison of at-site $T$-year event estimates of updated versus EC-generated IDFs shows at $T = $ 10-year, the return period commonly used for urban drainage design, current design standards require updates up
to 7% to mitigate the risk of urban flooding. Meanwhile, for longer recurrence interval ($T = $ 50-year or more), typical for critical civil infrastructural design, comparison of updated versus EC-generated IDF curves shows a difference ranges between 2% and 44% based on locations. These findings pose an important question: does the presence of nonstationarities in rainfall extremes require the design of nonstationary IDF curves? We argue that although it is crucial to recognize nonstationarity in precipitation extremes, the stationary form of IDFs can still
represent the extreme rainfall statistics for the present-day climate over Southern Ontario region. Our results are consistent with (Yilmaz et al., 2014; Yilmaz and Perera, 2013), in which authors found despite the presence of (statistically) significant trends in rainfall extremes; nonstationary GEV models did not show any additional advantages over the stationary models. As supported by the previous study (Singh et al., 2016), we attribute that the little or no changes in extreme rainfall statistics in the urbanized setting is due to the stabilization of urban
development leading to no substantial variations in the land use pattern. Hence, no significant changes in synoptic scale circulations, which in turn affect space-time pattern in rainfall extremes (Moglen and Schwartz, 2006).

Preliminary investigations based on regional and global climate model simulations in the study area confirm a considerable uncertainty in the projection of short-duration and high-intensity extreme rain events (Coulibaly et

al., 2015). While short-duration precipitation extremes are typically controlled by synoptic scale moisture convergence (Ruiz-Villanueva et al., 2012; Westra and Sisson, 2011), the daily extremes are often modulated by large-scale circulation patterns and local orographic factors (Carvalho et al., 2002; Gershunov and Barnett, 1998; Trenberth, 1999). The role of natural variability and multidecadal modes of sea-surface temperature (SST) in modulating Canadian extreme rainfall intensity have already been shown in the past (Gan et al., 2007; Shabbar et al., 1997). Further, review of the literature suggests that heavy precipitation does not necessarily lead to high stream discharge (Ivancic and Shaw, 2015; Do et al., 2017; Wasko and Sharma, 2017). The analysis of Do et al. (2017) reveals the trend in streamflow is more consistent across continental scale and neither the anthropogenic activities such as the presence of dams nor the vegetation cover have any significant effect on the results of trend estimates. Interestingly, the consensus among all three studies (Ivancic and Shaw, 2015; Do et al., 2017; Wasko and Sharma, 2017) is that the catchment size, which regulates the flow response because of antecedent moisture content, is the most important contributing factor in modulating the nature of trend in stream discharge. The smaller (especially, urban) catchments may have increased flood peaks, in contrast, larger (agricultural and rural) catchments may experience decreased runoff due to lower soil moisture. This can be attributed to the fact that high temperature leads to drying up of soil more quickly in larger catchments, thus forcing a large portion of precipitation not to become an overland flow. Finally, using more than 9000 daily streamflow records globally, Do et al. (2017) showed more stations with significant decreasing trends in annual maximum streamflow than that of significant increasing trends, indicating a limited evidence of increasing flood hazards. Their findings are corroborated with (Wasko and Sharma, 2017), in which authors showed that only in most extreme cases, for small catchments, increase in precipitation at higher temperature leads to increase in streamflow.

The statistical uncertainty in modeling nonstationarity can result from multiple sources. For instance, extrapolating the effects detected with observed historical series to the more extreme values that have not yet been experienced, model choices resulting from selection of covariates in the nonstationary distributions (Agilan and Umamahesh, 2017), and the treatment of nonstationarity introduced through either a linear (Ali and Mishra, 2017; Cheng et al., 2014; Westra et al., 2012) or polynomial (Villarini et al., 2009) trend, or a change-point (Renard et al., 2013) in the model. The key question remains how to update design events in a nonstationary climate. This becomes further challenging after trends and change points are detected in hydrometric time series. To address climate change adaptation needs under nonstationarity and uncertainty, some of the concepts discussed in the literature are design life level (Rootzén and Katz, 2013) to quantify the probability of exceeding a fixed threshold during the design life

of a project, replacing the commonly used concept of average return period with reliability (Read and Vogel, 2015) and a risk-based decision-making approach integrating the concept of *expected regret* (Rosner et al., 2014). However, apart from statistical uncertainty, one of the important sources of uncertainty in future planning period is the use of climate model output. Modeling nonstationarity in the future time period is complicated by the choice

of spatial resolution of climate models, lack of understanding of model physics due to different model choices and inherent uncertainties in climate model realizations resulting from different initial conditions, which is especially apparent over regional scales and decadal planning horizon (Ganguli et al., 2017; Hawkins and Sutton, 2009; Kumar and Ganguly, 2017; Meehl et al., 2009).

The increase in design storm during update process could also indicate a tendency towards an increase in mean precipitation and (or) a shift in the distribution, affecting its tail behavior. However, a few caveat remains, for example, a critical question could be: does an increase in DSI potentially linked towards more frequent and more intense precipitation extremes or is it an artifact of the new dataset in the update process? It is worthwhile to note that results shown here are manifestations of present-day climate using ground-based hydrometeorological

observations and the specific insights are nonetheless subject to the quality of available rainfall records. It remains an open-ended question to what extent we credibly develop IDFs in a changing climate (Coulibaly et al., 2015) since there is no uniformly accepted method of generating IDF information and related projection uncertainty in light of climate change. In general, highlighting advantages and limitations of nonstationary versus the stationary methods of analyses (Koutsoyiannis and Montanari, 2015; Montanari and Koutsoyiannis, 2014; Serinaldi and

Kilsby, 2015) is beyond the scope of the current study. Secondly, one of the limitations of the present analysis includes the lack of accounting for the consistency of the IDF curves in terms of shape enforcement. The lack of shape enforcement in the IDF curves (Figures 8 and S15) is the result of fitting separate distributions to the series of different durations. Finally, although in several instances, we find evidence of step changes in short-duration rainfall extremes, we have not introduced any change point model in the GEV parameters (Renard et al., 2013).

Future research should include two aspects. First, investigation of physical drivers (such as temperature, decadal and multidecadal modes of SST) influencing short-duration rainfall extremes, and inclusion of these covariates in nonstationary IDF development. Second, modeling nonstationarity by introducing a step-change model in GEV location and scale parameters. Finally, given that the findings reported herein, are for the current period (e.g.,

historical extreme rainfall time series), we recommend a careful extrapolation of these findings with regards to

future climate projections, in which frequency and magnitude of extreme rainfall are expected to intensify (Mailhot et al., 2012; Deng et al., 2016; Fischer and Knutti, 2016; Prein et al., 2016; Pfahl et al., 2017; Switzman et al. 2017). Further work should consider nonstationary methods for deriving future IDFs in Southern Ontario.

**Author Contribution**

P. Ganguli and P. Coulibaly designed the experiment. P. Ganguli carried out the experiment. P. Ganguli prepared the manuscript with contributions from P. Coulibaly.

**Acknowledgments**

The annual maximum rainfall data used in this study is downloaded from Environment and Climate Change Canada website: ftp://ftp.tor.ec.gc.ca/Pub/Engineering_Climate_Dataset/IDF/. Hourly and daily rainfall data are obtained
from Toronto and Region Conservation Authority (TRCA; https://trca.ca/) and Environment Canada Historical Climate Data Archive (http://climate.weather.gc.ca/). This work was supported by the Natural Science and Engineering Research Council (NSERC) Canadian FloodNet (Grant number: NETGP 451456). The first author of the manuscript would like to thank Dr. Jonas Olsson of Swedish Meteorological and Hydrological Institute (SMHI), Norrköping, Sweden for sharing MATLAB-based random cascade disaggregation tools and implementation details
through email. While change point and nonstationarity analyses were conducted in the statistical software R version 3.30 with add-on packages "trend", "fractal" and "cpm", the remaining analyses were performed in a MATLAB platform (MATLAB R2016a). The nonstationary GEV analyses were performed using MATLAB-based NEVA toolbox, available at the University of California, Irvine website: http://amir.eng.uci.edu/neva.php (as accessed on May 2016). The work was completed and communicated when the first author was a postdoctoral research fellow
at McMaster Water Resources and Hydrologic Modeling Lab, McMaster University, Canada.

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

## Figure Captions

**Figure 1**. (**a**) Selected urbanized sites in Southern Ontario. The Southern Ontario (41° - 44°N, 84° - 76°W) is the southernmost region of Canada and is situated on a southwest-northeast transect, bounded by lakes Huron, Erie, and Ontario. The nine locations on the map are (from southwest to northeast corner): Windsor Airport, London International Airport, Stratford Wastewater Treatment Plant (WWTP), Fergus Shand Dam, Hamilton Airport, Toronto  International Airport, Oshawa Water Pollution Control Plant (WPCP), Trenton Airport, and Kingston Pumping Station. Topography map indicates maximum slope of 670 m above mean sea level. (b) The population map shows six the sites: Windsor Airport, London International Airport, Hamilton Airport, Toronto International Airport, Oshawa WPCP, and Kingston P. Station to be located either in or the vicinity of densely populated urbanized area. The remaining three sites are located in the moderately populated area. The daily and sub-daily AMP records in all locations vary between the minimum of 46 and maximum of 66 years.

**Figure 2**. Schematics of the process flow (Blue - input step, orange - process step, and green – decision steps). All three tests – Mann-Kendall, Pettitt's and Mann-Whitney, check for shifts in the mean. While Mann-Kendall tests for monotonic trends, the other two tests, Pettitt's and Mann-Whitney check for change point or regime shift in the time series.

**Figure 3**. Spatial distribution of trends, change points and nonstationarities in rainfall extremes of several durations in nine urbanized locations, Southern Ontario (**a – g**). The up and down triangles in white indicate (statistically insignificant) up and downward shifts; the up and down triangles in cyan and orange indicate shifts with change points only; the up and down triangles in the dark blue and red show presence of (statistically significant) trends including change point(s). A 'x' symbol in the triangle indicates nonstationarity detected through Priestley and Subbarao test statistics. All tests are performed at 10% significance levels, i.e., p-value < 0.10.

**Figure 4**. DSI estimates of median (horizontal line within the box plot) and 95% credible intervals for 100-year return periods of stationary versus nonstationary models across nine sites (**a - i**). The boxplots indicate the uncertainty in estimated DSI using Bayesian inference.

**Figure 5**. Percentage changes (in top panel) and Z-statistics (bottom panel) of at-site T-year event estimates for T = 2-year to T = 100-year return periods (a – d) with durations between 15-min and 24-hr in nine urbanized locations, Southern Ontario. The Z-statistic represents statistical significance of differences in DSI obtained from the best

selected nonstationary versus the stationary model. The Z-statistics is statistically significant when $|Z| > 1.64$ at 10% significance level. The shades in blue and red denote decrease and increase in Z-statistics with the strength of shading represents the magnitude of the test statistics. The cyan shading represents the site with significant autocorrelation, which we exclude from the analysis.

**Figure 6**. Central tendency (median, **b**) and the bounds (95% credible interval, **a** and **c**) of the updated nonstationary versus EC-generated T = 2-and 10-year event estimates for DSI at selected return periods with durations between 15-min and 24-hr. The DSI and associated 95% confidence limits of EC-generated IDF is obtained from the national archive of Engineering Climate Datasets (http://climate.weather.gc.ca/). The shades in blue and red denote decrease and increase in DSI. The strength of shading represents the magnitude of the ratio between updated versus EC-generated DSI.

**Figure 7.** Central tendency (median, **b**) and the bounds (95% credible interval, **a** and **c**) of the updated nonstationary versus EC-generated T = 50-and 100-year event estimates for DSI at selected return periods with durations between 15-min and 24-hr. The DSI and associated 95% confidence limits of EC-generated IDF is obtained from the national archive of Engineering Climate Datasets (http://climate.weather.gc.ca/). The shades in blue and red denote decrease and increase in DSI. The strength of shading represents the magnitude of the ratio between updated versus EC-generated DSI.

**Figure 8.** Estimated nonstationary versus EC-generated IDFs for T = 2, 5, 10, 25, 50 and 100-year return periods for the selected urbanized locations in Southern Ontario, Canada.

**Table 1**. Selected station locations, population distribution and hourly and daily data availability

| Stations | EC-Station ID | Lat (°) | Long (°) | Elevation (m) | Population Estimate | Census Subdivision | EC-derived AMP | Hourly Rainfall | Daily Rainfall | Missing years/ Duration values |
|---|---|---|---|---|---|---|---|---|---|---|
| Toronto P. Int'l Airport | 6158731 | 43.68 | -79.63 | 173.4 | 5,583,046 | Toronto CMA[1] | 1950 - 2013 | 1960 - 2012 | 1940 - 2013 | 1952-53, 2005 [15-30 min, 1-6 hr] |
| Hamilton Airport | 6153194 | 43.17 | -79.93 | 237.7 | 519,949 | Population Center | 1971 - 2003 | 1971 - 2003 | 1960 – 2010 | 2004-2010 |
| Oshawa WPCP | 6155878 | 43.87 | -78.83 | 83.8 | 356,177 | Oshawa CMA | 1970 - 2006 | 1970 – 1999 | 1970 - 2015 | 1971 [15-30 min, 1-6hr], 1995 [12 hr], 1999 [6-12hr], 2000, 2005-06 [15-30 min, 1-6 hr], 2007-15 |
| Windsor Airport | 6139525 | 42.28 | -82.96 | 189.6 | 319,246 | Windsor CMA | 1946 - 2007 | 1960 – 2007 | 1940 - 2013 | 1950, 2008 - 2013 |
| Kingston P. Station | 6104175 | 44.24 | -76.48 | 76.5 | 159,561 | Kingston CMA | 1961 - 2007 | 1961 – 2003 | 1960 – 2007 | 2004 |
| London Int'l Airport | 6144478/75[*] | 43.03 | -81.15 | 278 | 474,786 | London CMA | 1950 - 2007[*] | 1961 – 2001 | 1940 – 2015 | 1950-51, 2002, 2008-2015 |
| Trenton Airport | 6158875 | 44.12 | -77.53 | 86.3 | 43,086 | Population Center | 1965 - 2013 | 1964 – 1997 | 1935 – 2015 | 1974, 1998-99, 2002 [15-30 min, 1-6hr], 2003-04 |
| Stratford WWTP | 6148100 | 43.37 | -81.0 | 345 | 30,886 | CA | 1966 - 2004 | 1966 – 2007 | 1960 – 2015 | 1973, 1999 |
| Fergus Shand Dam | 6142400 | 43.73 | -80.33 | 417.6 | 19,126 | Population Center | 1961 - 2007 | 1960 – 2007 | 1950 – 2015 | 1969, 1971, 1986, 1987 [2- 6hr], 1992 [6 - 12hr], 1995 |

[1] CMA and CA denote census metropolitan area and census agglomeration respectively. Statistics of Canada defines a CMA with a population density of at least 100,000, where the urban core of that area has at least 50,000 people, whereas CA must have an urban core population density of at least 10,000. A population Center (or urban area) is an area with at least a population of 1,000 and a density of 400 or more people per square kilometer. All population information are collected from Statistics Canada (https://www12.statcan.gc.ca/) website.[*]Missing values are infilled using observations from nearest Environment Canada station ID 6144475 (latitude 44° and longitude -81.5°) located at 111.5 km geodesic distance. Annual maxima values of missing years or durations are obtained by disaggregating daily data to hourly and sub-hourly time steps.

**Table 2**. Performance of stationary and nonstationary models for Toronto Pearson International Airport

| Time Slice | Model | Location parameter | Scale parameter | Shape parameter | AIC$_c$ | Bayes-factor | LB (100yr) | UB (100yr) | UB/LB |
|---|---|---|---|---|---|---|---|---|---|
| 15-min | GEV$_t$-0 | 37.02 | 19.83 | -0.073 | -465.05 | - | 78.85 | 209.56 | 2.66 |
| | GEV$_t$-I | 30.11 + 0.194$t$ | 20.86 | -0.079 | **-450.28** | 4.98 | 105.3 | 229.77 | 2.18 |
| | GEV$_t$-II | 34.30 + 0.056$t$ | exp(2.68 + 0.0069$t$) | -0.11 | -383.58 | 10.47 | 87.66 | 119.42 | 1.36 |
| 30-min | GEV$_t$-0 | 25.65 | 13.14 | 0.019 | -442.29 | - | 58.8 | 155.85 | 2.65 |
| | GEV$_t$-I | 17.32 + 0.21$t$ | 13.44 | -0.075 | **-422.67** | 1.47 | 57.9 | 113.37 | 1.96 |
| | GEV$_t$-II | 12.08 + 0.35$t$ | exp(2.77 + 0.0023$t$) | -0.20 | -351.22 | 74357.2 | 58.63 | 99 | 1.69 |
| 1-hr | GEV$_t$-0 | 19.77 | 7.79 | 0.07 | -477.68 | - | 45.47 | 101.33 | 2.22 |
| | GEV$_t$-I | 18.27 + 0.022$t$ | 8.59 | -0.08 | **-402.43** | 78.53 | 42.27 | 63.57 | 1.50 |
| | GEV$_t$-II | 4.44 + 0.414$t$ | exp(1.71 + 0.015$t$) | 0.044 | -372.2 | 4.43×10$^9$ | 49.65 | 87.11 | 1.75 |
| 2-hr | GEV$_t$-0 | 11.79 | 4.45 | 0.11 | -477.64 | - | 28.24 | 58.52 | 2.07 |
| | GEV$_t$-I | 11.0 + 0.02$t$ | 4.74 | -0.02 | **-449.02** | 13.95 | 27.24 | 40.98 | 1.50 |
| | GEV$_t$-II | 11.46 – 0.0053$t$ | exp(1.52 – 0.00072$t$) | 0.28 | -421.44 | 9.08 | 46.44 | 61.47 | 1.32 |
| 6-hr | GEV$_t$-0 | 4.98 | 1.50 | 0.26 | -488.39 | - | 13.71 | 21 | 1.53 |
| | GEV$_t$-I | 5.12+0.0005$t$ | 1.57 | 0.24 | **-496.92** | 0.15 | 12.02 | 29.77 | 2.48 |
| | GEV$_t$-II | 5.44-0.0049$t$ | exp(0.77 – 0.0042$t$) | 0.10 | -424.07 | 52.13 | 13.71 | 21.0 | 1.53 |
| 12-hr | GEV$_t$-0 | 2.96 | 0.70 | 0.36 | -503.23 | - | 6.59 | 25.72 | 3.90 |
| | GEV$_t$-I | 3.02-0.0031$t$ | 0.69 | 0.51 | -501.42 | 1.39 | 12.4 | 21.98 | 1.77 |
| | GEV$_t$-II | 3.13-0.0045$t$ | exp(-0.183-0.0032$t$) | 0.49 | **-511.69** | 0.86 | 12.89 | 20.58 | 1.60 |
| 24-hr | GEV$_t$-0 | 1.71 | 0.41 | 0.29 | -477.04 | - | 3.69 | 11.71 | 3.17 |
| | GEV$_t$-I | 1.73-0.0006$t$ | 0.41 | 0.28 | **-466.25** | 13.22 | 3.75 | 10.41 | 2.78 |
| | GEV$_t$-II | 1.66+0.00093$t$ | exp(-1.00+0.00274$t$) | 0.30 | -460.06 | 1.30 | 4.28 | 8.16 | 1.91 |

[*]GEV$_t$-0 is stationary model whereas GEV$_t$-I and GEV$_t$-II are nonstationary models with time-variant mean, and both time-variant mean and standard deviation respectively. The selected best fitted nonstationary model is marked in bold letters. Bayes factor, $\gamma < 1$ indicates that the nonstationary model fits better than the stationary model. However, in cases $\gamma > 1$, to compare with stationary model, the nonstationary model is selected following minimum *AICc* criteria. LB and UB indicate lower and upper bounds of DSI at 100-year return period.

**Table 3**. Performance of stationary and nonstationary models for Hamilton Airport

| Time Slice | Model | Location parameter | Scale parameter | Shape parameter | $AIC_c$ | Bayes-factor | LB (100yr) | UB (100yr) | UB/LB |
|---|---|---|---|---|---|---|---|---|---|
| 15-min | $GEV_t$-0 | 53.84 | 14.96 | 0.12 | -347.44 | - | 103.8 | 221.98 | 2.14 |
| | $GEV_t$-I | 56.31-0.096$t$ | 14.2 | 0.14 | **-338.32** | 0.67 | 102.19 | 223.91 | 2.19 |
| | $GEV_t$-II | 55.86-0.114$t$ | exp(2.83-0.0056$t$) | 0.19 | -351.58 | 2.07 | 107.81 | 285.01 | 2.64 |
| 30-min | $GEV_t$-0 | 27.1 | 7.32 | 0.20 | -369.40 | - | 56.71 | 174.66 | 3.08 |
| | $GEV_t$-I | 28.002-0.06$t$ | 7.28 | 0.11 | -346.29 | 1.73 | 53.93 | 99.27 | 1.84 |
| | $GEV_t$-II | 27.8-0.038$t$ | exp(1.91+0.0009$t$) | 0.25 | **-365.19** | 0.28 | 69.81 | 110.76 | 1.59 |
| 1-hr | $GEV_t$-0 | 21.79 | 6.41 | 0.13 | -361.35 | - | 41.92 | 109.07 | 2.60 |
| | $GEV_t$-I | 21.33+0.026$t$ | 7.06 | 0.03 | **-353.4** | 0.62 | 45.85 | 75.54 | 1.65 |
| | $GEV_t$-II | 20.50+0.046$t$ | exp(1.86+0.0035$t$) | -0.0039 | -350.97 | 3.09 | 43.75 | 68.8 | 1.57 |
| 2-hr | $GEV_t$-0 | 12.63 | 3.68 | 0.11 | -349.70 | - | 25.81 | 51.37 | 1.99 |
| | $GEV_t$-I | 12.15+0.006$t$ | 3.76 | 0.21 | -322.00 | 4.91 | 32.16 | 54.78 | 1.70 |
| | $GEV_t$-II | 11.53+0.042$t$ | exp(1.09+0.0087$t$) | 0.19 | **-329.09** | 11.20 | 32.76 | 49.51 | 1.51 |
| 6-hr | $GEV_t$-0 | 5.32 | 1.33 | 0.23 | -389.88 | - | 10.24 | 32.46 | 3.17 |
| | $GEV_t$-I | 5.15+0.0037$t$ | 1.28 | 0.29 | **-396.75** | 0.94 | 14.51 | 21.47 | 1.48 |
| | $GEV_t$-II | 5.09+0.0052$t$ | exp(0.12+0.0038$t$) | 0.28 | -375.03 | 1.21 | 14.53 | 20.47 | 1.41 |
| 12-hr | $GEV_t$-0 | 3.11 | 0.74 | 0.20 | -369.54 | - | 5.86 | 15.58 | 2.66 |
| | $GEV_t$-I | 3.09+0.0022$t$ | 0.74 | 0.27 | **-366.46** | 1.73 | 14.51 | 21.47 | 1.48 |
| | $GEV_t$-II | 3.03+0.0023$t$ | exp(-0.305+0.0002$t$) | 0.21 | -363.03 | 1.12 | 13.97 | 6.35 | 2.20 |
| 24-hr | $GEV_t$-0 | 1.44 | 0.49 | 0.16 | -338.35 | - | 3.05 | 11.47 | 3.76 |
| | $GEV_t$-I | 1.36+0.0026$t$ | 0.48 | 0.22 | **-338.33** | 0.31 | 3.26 | 8.42 | 2.58 |
| | $GEV_t$-II | 1.33+0.0034$t$ | exp(-0.74-0.00019$t$) | 0.20 | -326.63 | 0.99 | 4.04 | 6.44 | 1.59 |

[*]$GEV_t$-0 is stationary model whereas $GEV_t$-I and $GEV_t$-II are nonstationary models with time-variant mean, and both time-variant mean and standard deviation respectively. The selected best fitted nonstationary model is marked in bold letters. Bayes factor, $\gamma < 1$ indicates that the nonstationary model fits better than the stationary model. However, in cases $\gamma > 1$, to compare with stationary model, the nonstationary model is selected following minimum *AICc* criteria. LB and UB indicate lower and upper bounds of DSI at 100-year return period.

**Table 4**. Performance of stationary and nonstationary models for Windsor Airport

| Time Slice | Model | Location parameter | Scale parameter | Shape parameter | $AIC_c$ | Bayes-factor | LB (100yr) | UB (100yr) | UB/LB |
|---|---|---|---|---|---|---|---|---|---|
| 15-min | $GEV_t$-0 | 60.04 | 15.76 | 0.13 | -394.2 | - | 106.43 | 300.24 | 2.82 |
| | $GEV_t$-I | 61.6-0.099$t$ | 14.61 | 0.25 | **-370.00** | 0.80 | 157.9 | 227.06 | 1.44 |
| | $GEV_t$-II | 63.33-0.068$t$ | exp(2.67+0.0027$t$) | 0.013 | -376.75 | 5.36 | 115.47 | 166.94 | 1.44 |
| 30-min | $GEV_t$-0 | 38.92 | 12.94 | 0.06 | -443.89 | - | 72.9 | 179.38 | 2.46 |
| | $GEV_t$-I | 43.20-0.124$t$ | 12.04 | 0.12 | **-435.12** | 0.25 | 72.6 | 210.06 | 2.89 |
| | $GEV_t$-II | 42.81-0.11$t$ | exp(2.33+0.0032$t$) | -0.0096 | -371.83 | 1.002 | 81.7 | 104.43 | 1.28 |
| 1-hr | $GEV_t$-0 | 24.82 | 8.00 | 0.044 | -459.27 | - | 46.8 | 112.2 | 2.40 |
| | $GEV_t$-I | 28.93-0.14$t$ | 7.1 | 0.14 | **-452.65** | 0.35 | 58.99 | 89.75 | 1.52 |
| | $GEV_t$-II | 28.86-0.12$t$ | exp(2.11-0.0005$t$) | 0.024 | -444.90 | 0.22 | 56.66 | 73.64 | 1.30 |
| 2-hr | $GEV_t$-0 | 15.79 | 5.12 | -0.14 | -476.71 | - | 25.62 | 45.09 | 1.76 |
| | $GEV_t$-I | 17.58-0.073$t$ | 4.78 | -0.02 | -434.62 | 0.38 | 26.13 | 58.27 | 2.23 |
| | $GEV_t$-II | 17.70-0.07$t$ | exp(1.50+0.0049$t$) | -0.14 | **-475.53** | 0.16 | 29.79 | 37.83 | 1.27 |
| 6-hr | $GEV_t$-0 | 6.24 | 1.86 | 0.041 | -472.16 | - | 11.85 | 23.23 | 1.96 |
| | $GEV_t$-I | 6.80-0.014$t$ | 1.94 | -0.04 | -477.91 | 0.52 | 12.45 | 16.77 | 1.35 |
| | $GEV_t$-II | 6.85-0.017$t$ | exp(0.64+0.0013$t$) | 0.040 | **-480.19** | 0.65 | 14.28 | 18.75 | 1.31 |
| 12-hr | $GEV_t$-0 | 3.47 | 0.98 | 0.10 | -489.97 | - | 6.3 | 16.67 | 2.65 |
| | $GEV_t$-I | 3.97-0.016$t$ | 0.92 | 0.14 | -461.74 | 0.20 | 6.75 | 14.51 | 2.15 |
| | $GEV_t$-II | 3.89-0.012$t$ | exp(-0.055+0.00094$t$) | 0.09 | **-481.94** | 0.20 | 7.7 | 10.57 | 1.37 |
| 24-hr | $GEV_t$-0 | 2.04 | 0.53 | 0.03 | -475.90 | - | 3.42 | 7.36 | 2.15 |
| | $GEV_t$-I | 2.05-0.011$t$ | 0.53 | 0.03 | **-472.08** | 2.44 | 3.41 | 7.15 | 2.09 |
| | $GEV_t$-II | 1.74+0.0067$t$ | exp(-0.78+0.0054$t$) | 0.0056 | -415.7 | 30.74 | 3.66 | 5.84 | 1.60 |

[*]$GEV_t$-0 is stationary model whereas $GEV_t$-I and $GEV_t$-II are nonstationary models with time-variant mean, and both time-variant mean and standard
5  deviation respectively. The selected best fitted nonstationary model is marked in bold letters. Bayes factor, $\gamma < 1$ indicates that the nonstationary
   model fits better than the stationary model. However, in cases $\gamma > 1$, to compare with stationary model, the nonstationary model is selected following
   minimum $AICc$ criteria. LB and UB indicate lower and upper bounds of DSI at 100-year return period.

**Table 5.** Performance of stationary and nonstationary models for London International Airport

| Time Slice | Model | Location parameter | Scale parameter | Shape parameter | AIC$_c$ | Bayes-factor | LB (100yr) | UB (100yr) | UB/LB |
|---|---|---|---|---|---|---|---|---|---|
| 15-min | GEV$_t$-0 | 51.7 | 19.15 | 0.045 | -449.07 | - | 112.5 | 206.6 | 1.84 |
|  | GEV$_t$-I | 57.8-0.12$t$ | 18.64 | 0.19 | -457.44 | 0.30 | 119.5 | 311.1 | 2.60 |
|  | GEV$_t$-II | 59.71-0.24$t$ | exp(2.95-0.00095$t$) | 0.17 | **-459.62** | 0.24 | 149.51 | 222.9 | 1.49 |
| 30-min | GEV$_t$-0 | 34.04 | 11.26 | 0.16 | -535.35 | - | 70.04 | 264.2 | 3.77 |
|  | GEV$_t$-I | 35.42 − 0.082$t$ | 11.88 | 0.054 | -433.14 | 9.61 | 68.83 | 142.8 | 2.07 |
|  | GEV$_t$-II | 38.42 − 0.13$t$ | exp(2.39-0.00304$t$) | 0.12 | **-446.35** | 0.09 | 74.91 | 125.6 | 1.68 |
| 1-hr | GEV$_t$-0 | 19.06 | 6.92 | 0.14 | -511.32 | - | 44.78 | 110.3 | 2.46 |
|  | GEV$_t$-I | 20.5-0.042$t$ | 6.75 | 0.21 | **-511.94** | 2.26 | 57.91 | 95.6 | 1.65 |
|  | GEV$_t$-II | 25.2-0.18$t$ | exp(2.68-0.0194$t$) | 0.052 | -494.92 | 206.09 | 40.8 | 116.9 | 2.87 |
| 2-hr | GEV$_t$-0 | 11.93 | 4.57 | 0.046 | -501.85 | - | 26.04 | 56.4 | 2.17 |
|  | GEV$_t$-I | 13.29-0.044$t$ | 4.49 | 0.093 | **-496.55** | 1.32 | 30.05 | 47.1 | 1.57 |
|  | GEV$_t$-II | 12.60-0.029$t$ | exp(1.42+0.00196$t$) | 0.20 | -462.71 | 1.27 | 37.68 | 54.6 | 1.45 |
| 6-hr | GEV$_t$-0 | 5.196 | 1.47 | 0.082 | -498.12 | - | 9.79 | 19.9 | 2.03 |
|  | GEV$_t$-I | 5.80-0.018$t$ | 1.35 | 0.058 | **-501.40** | 0.05 | 10.48 | 14.5 | 1.38 |
|  | GEV$_t$-II | 5.83-0.018$t$ | exp(0.32-0.0012$t$) | 0.099 | -499.38 | 0.02 | 10.14 | 19.2 | 1.89 |
| 12-hr | GEV$_t$-0 | 3.09 | 0.80 | -0.0013 | -515.05 | - | 5.35 | 10.1 | 1.89 |
|  | GEV$_t$-I | 3.34-0.008$t$ | 0.80 | 0.062 | **-511.60** | 0.13 | 6.32 | 8.7 | 1.38 |
|  | GEV$_t$-II | 3.49-0.011$t$ | exp(-0.22-0.002$t$) | -0.026 | -500.35 | 0.05 | 5.72 | 7.5 | 1.30 |
| 24-hr | GEV$_t$-0 | 1.72 | 0.63 | -0.051 | -473.40 | - | 3.14 | 6.3 | 2.01 |
|  | GEV$_t$-I | 1.98-0.008$t$ | 0.61 | -0.054 | **-450.07** | 0.17 | 3.57 | 5.0 | 1.41 |
|  | GEV$_t$-II | 2.036-0.008$t$ | exp(-0.45-0.0007$t$) | -0.103 | -435.8 | 0.12 | 3.44 | 4.9 | 1.44 |

[*]GEV$_t$-0 is stationary model whereas GEV$_t$-I and GEV$_t$-II are nonstationary models with time-variant mean, and both time-variant mean and standard deviation respectively. The selected best fitted nonstationary model is marked in bold letters. Bayes factor, $\gamma < 1$ indicates that the nonstationary model fits better than the stationary model. However, in cases $\gamma > 1$, to compare with stationary model, the nonstationary model is selected following minimum *AICc* criteria. LB and UB indicate lower and upper bounds of DSI at 100-year return period.

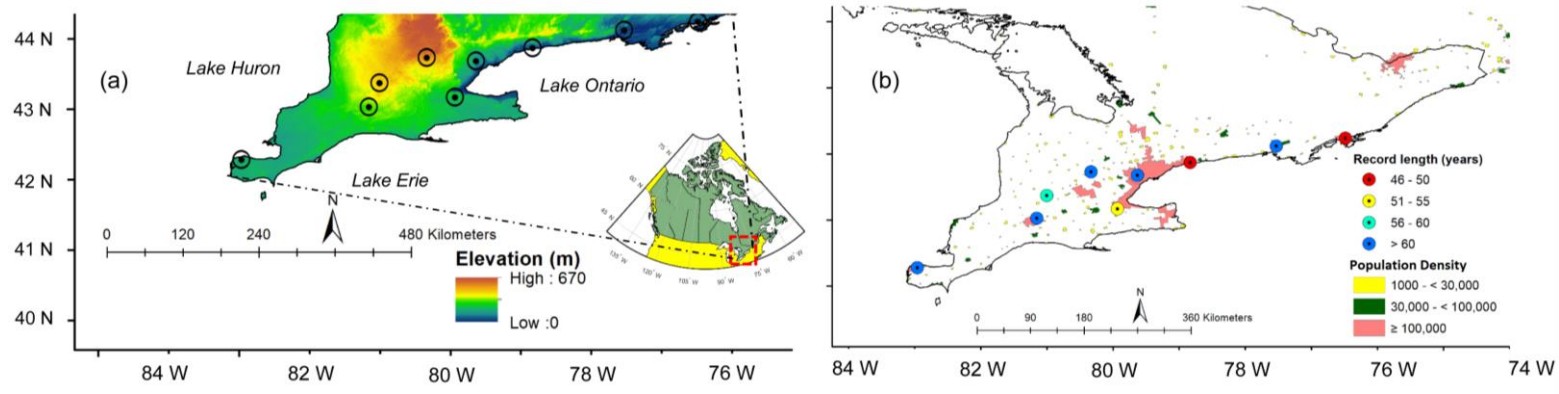

**Figure 1**. (**a**) Selected urbanized sites in Southern Ontario. The Southern Ontario (41° - 44°N, 84° - 76°W) is the southernmost region of Canada and is situated on a southwest-northeast transect, bounded by lakes Huron, Erie, and Ontario. The nine locations on the map are (*from southwest to northeast corner*): Windsor Airport, London International Airport, Stratford Wastewater Treatment Plant (WWTP), Fergus Shand Dam, Hamilton Airport, Toronto International Airport, Oshawa Water Pollution Control Plant (WPCP), Trenton Airport, and Kingston Pumping Station. Topography map indicates the maximum slope of 670 m above mean sea level. (**b**) The population map shows six the sites: Windsor Airport, London International Airport, Hamilton Airport, Toronto International Airport, Oshawa WPCP, and Kingston P. Station to be located either in or the vicinity of densely populated urbanized area. The remaining three sites are located in the moderately populated area. The short-duration AMP records in all locations vary between the minimum of 46 and maximum of 66 years.

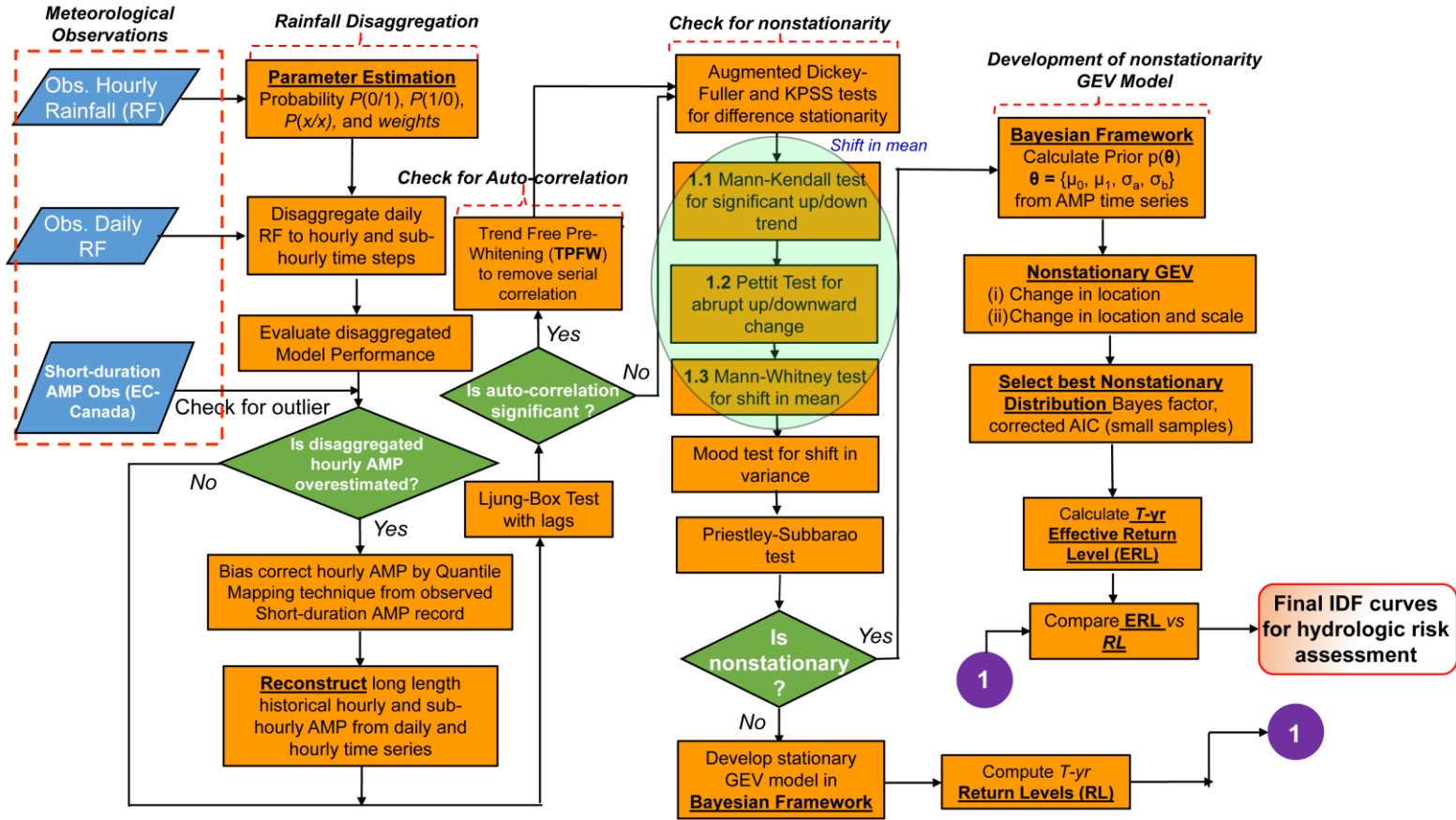

**Figure 2**. Schematics of the process flow (Blue - input step, orange - process step, and green – decision steps). All three tests – Mann-Kendall, Pettitt's and Mann-Whitney, check for shifts in the mean. While Mann-Kendall tests for monotonic trends, the other two tests, Pettitt's and Mann-Whitney check for change point or regime shift in the time series.

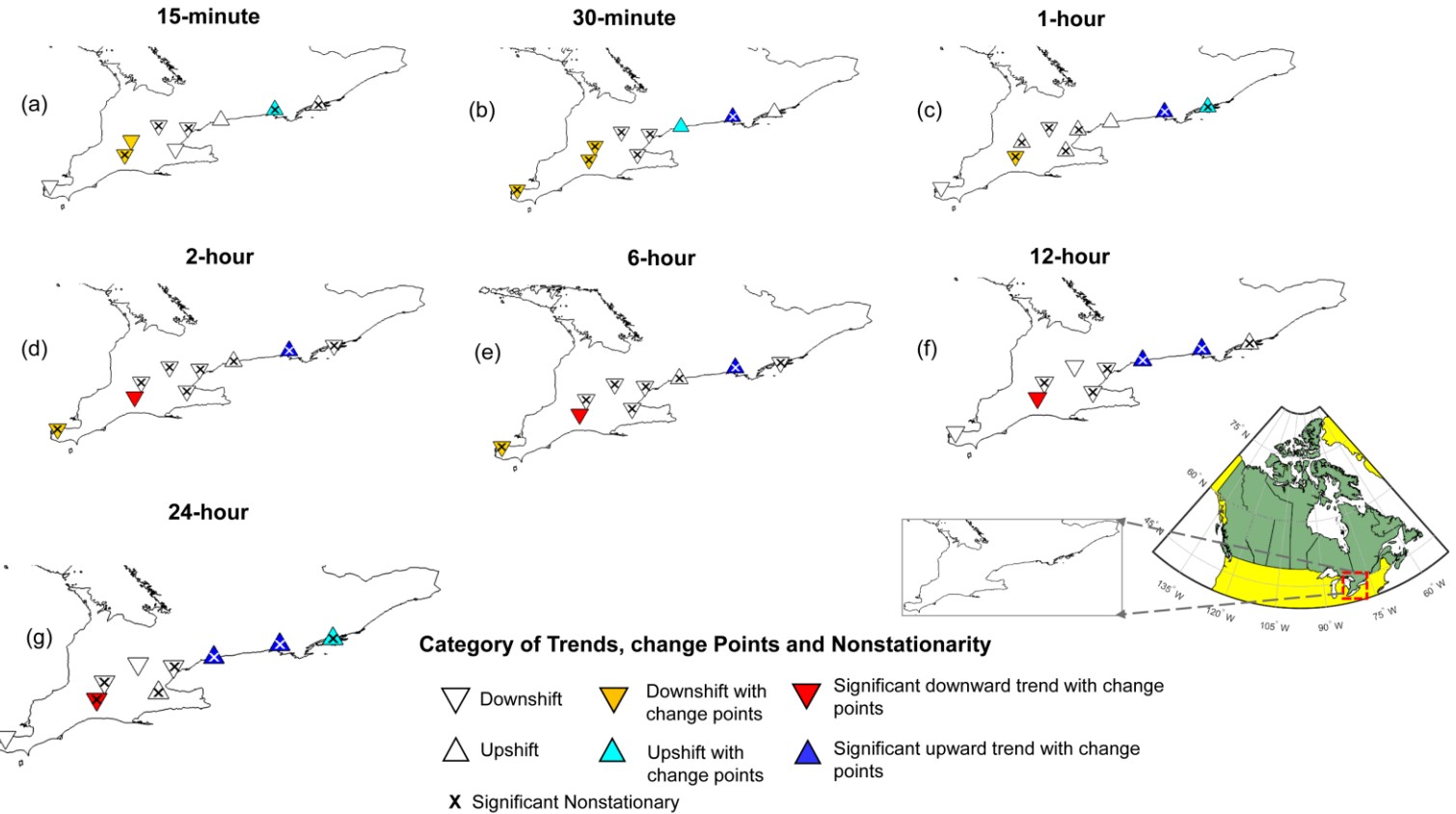

**Figure 3**. Spatial distribution of trends, change points and nonstationarities in rainfall extremes of several durations in nine urbanized locations, Southern Ontario (**a – g**). The up and down triangles in white indicate (statistically insignificant) up and downward shifts; the up and down triangles in cyan and orange indicate shifts with change points only; the up and down triangles in the dark blue and red show presence of (statistically significant) trends including change point(s). A 'x' symbol in the triangle indicates nonstationarity detected through Priestley and Subbarao test statistics. All tests are performed at 10% significance levels, i.e., p-value < 0.10.

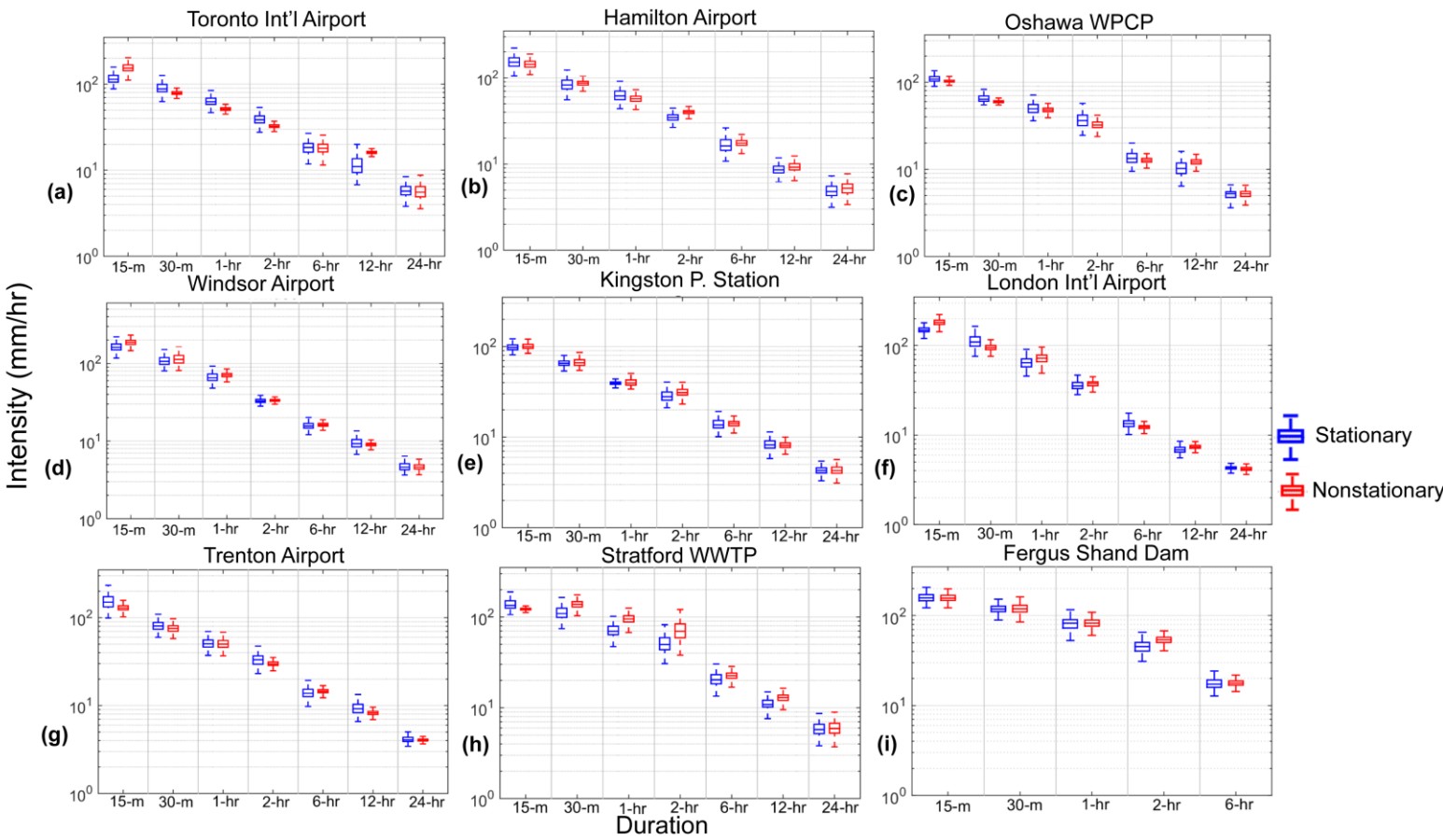

**Figure 4**. DSI estimates of the median (horizontal line within the box plot) and 95% credible intervals for 100-year return periods of stationary versus nonstationary models across nine sites (**a - i**). The boxplots indicate the uncertainty in estimated DSI using Bayesian inference.

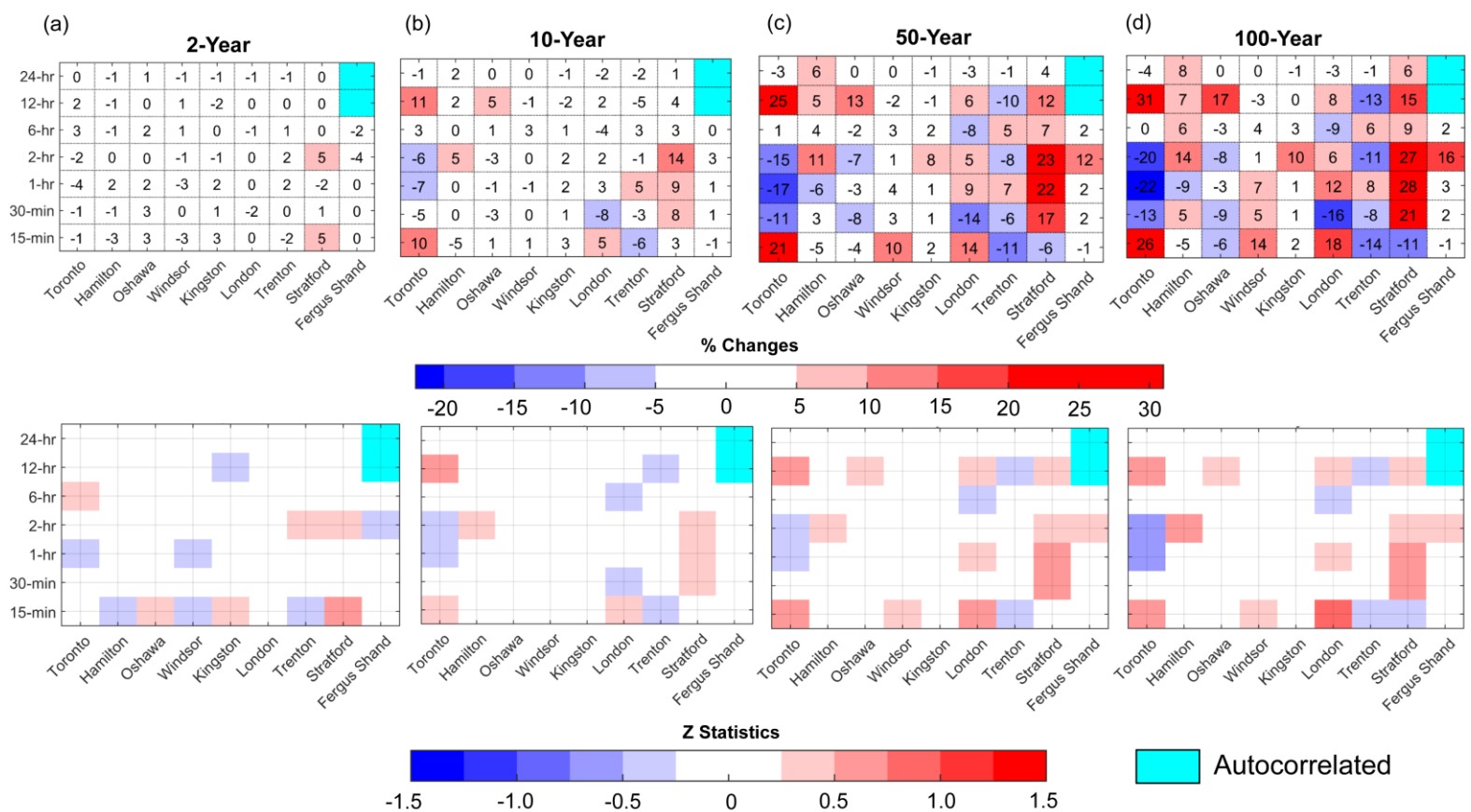

**Figure 5**. Percentage changes (in top panel) and Z-statistics (bottom panel) of at-site $T$-year event estimates for $T$ = 2-year to $T$ = 100-year return periods (**a** – **d**) with durations between 15-min and 24-hr in nine urbanized locations, Southern Ontario. The Z-statistic represents statistical significance of differences in DSI obtained from the best selected nonstationary versus the stationary model. The Z-statistics is statistically significant when |Z| > 1.64 at 10% significance level. The shades in blue and red denote decrease and increase in Z-statistics with the strength of shading represents the magnitude of the test statistics. The durations with significant autocorrelations are excluded from the analysis.

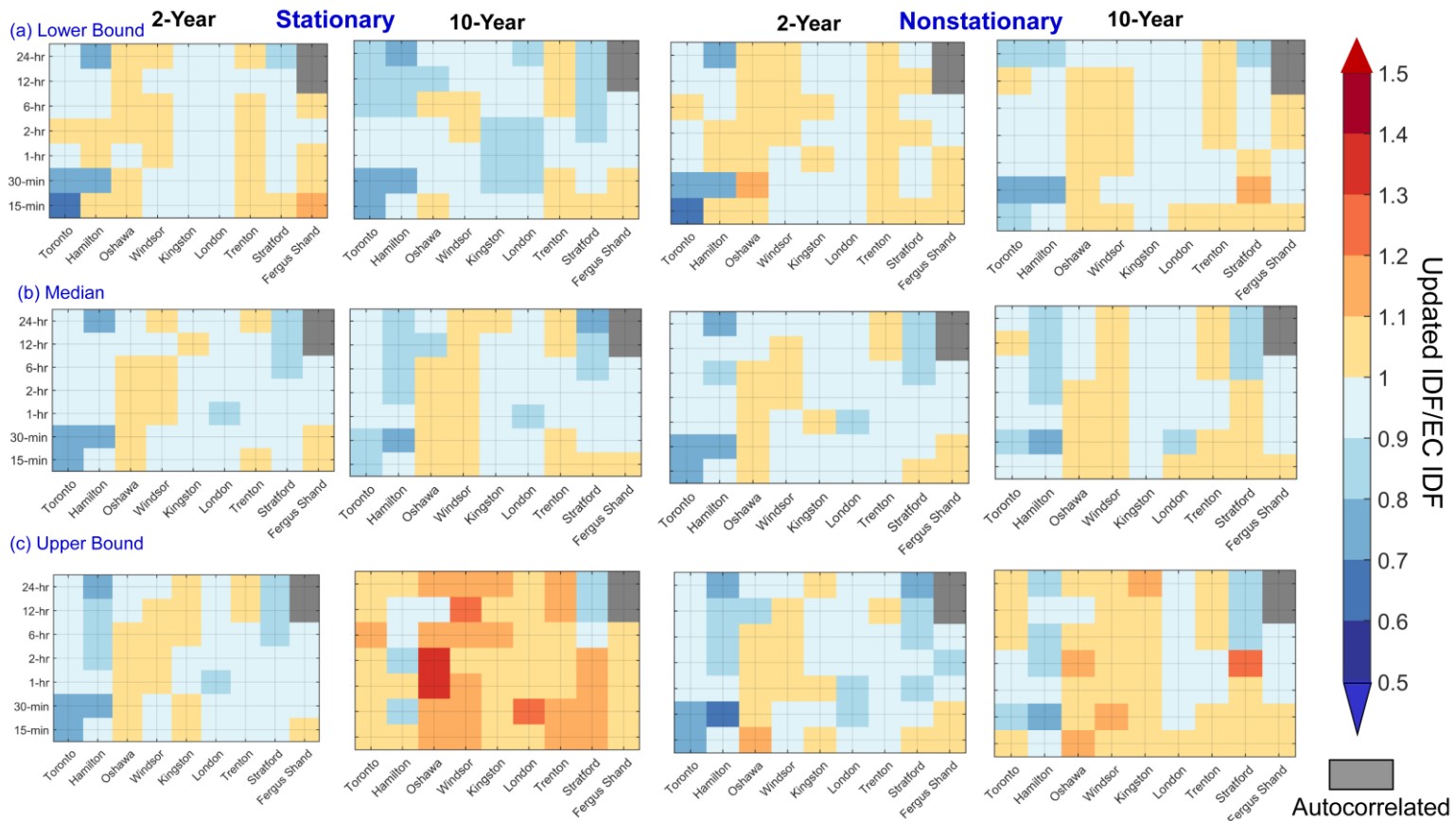

**Figure 6**. Central tendency (median, **b**) and the bounds (95% credible interval, **a** and **c**) of the updated nonstationary versus EC-generated $T$ = 2- and 10-year event estimates for DSI at selected return periods with durations between 15-min and 24-hr. The DSI and associated 95% confidence limits of EC-generated IDF is obtained from the national archive of Engineering Climate Datasets (http://climate.weather.gc.ca/). The shades in blue and red denote decrease and increase in DSI. The strength of shading represents the magnitude of the ratio between updated versus EC-generated DSI.

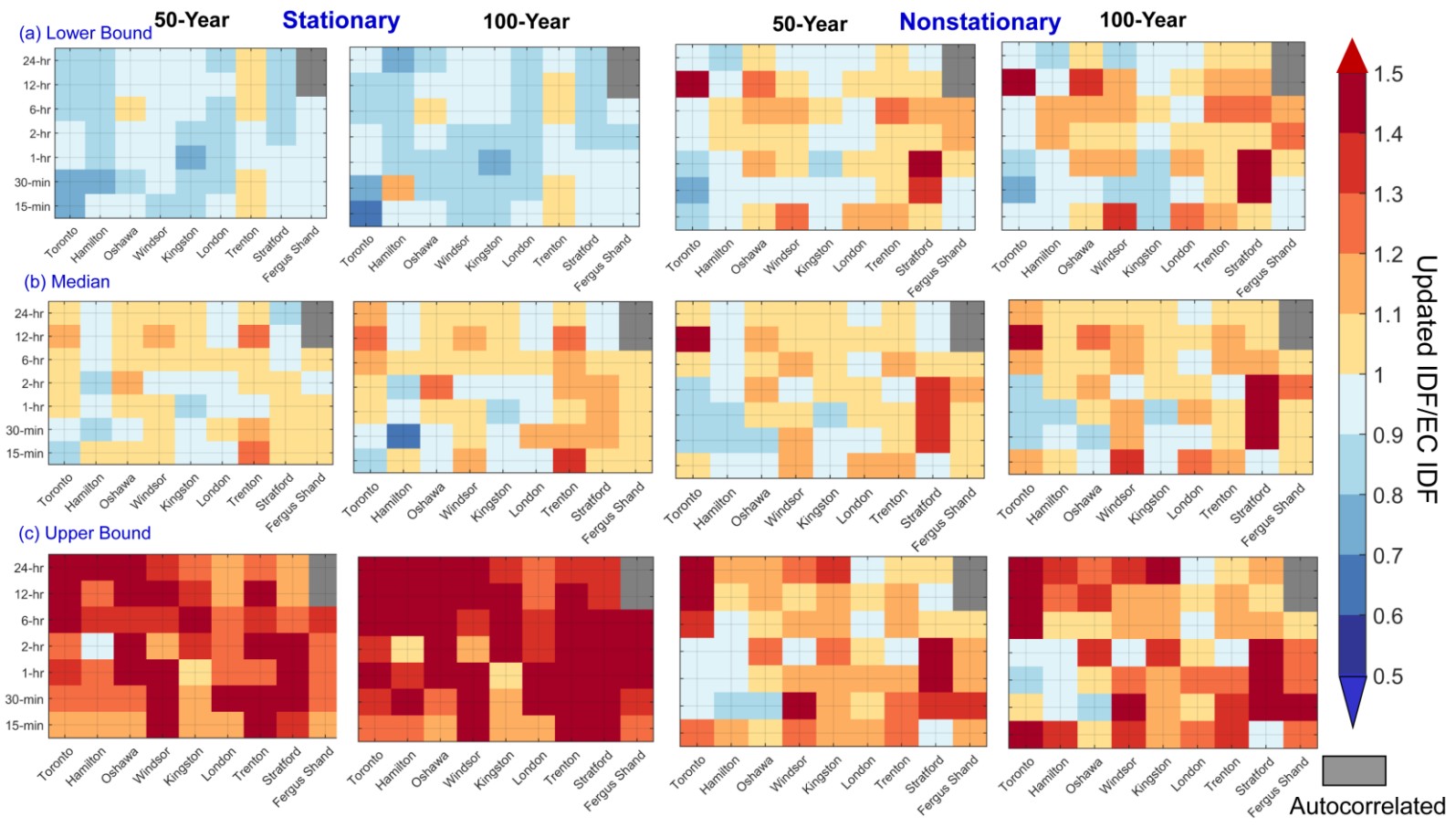

**Figure 7.** Central tendency (median, b) and the bounds (95% credible interval, **a** and **c**) of the updated nonstationary versus EC-generated T = 50-and 100-year event estimates for DSI at selected return periods with durations between 15-min and 24-hr. The DSI and associated 95% confidence limits of EC-generated IDF is obtained from the national archive of Engineering Climate Datasets (http://climate.weather.gc.ca/). The shades in blue and red denote decrease and increase in DSI. The strength of shading represents the magnitude of the ratio between updated versus EC-generated DSI.

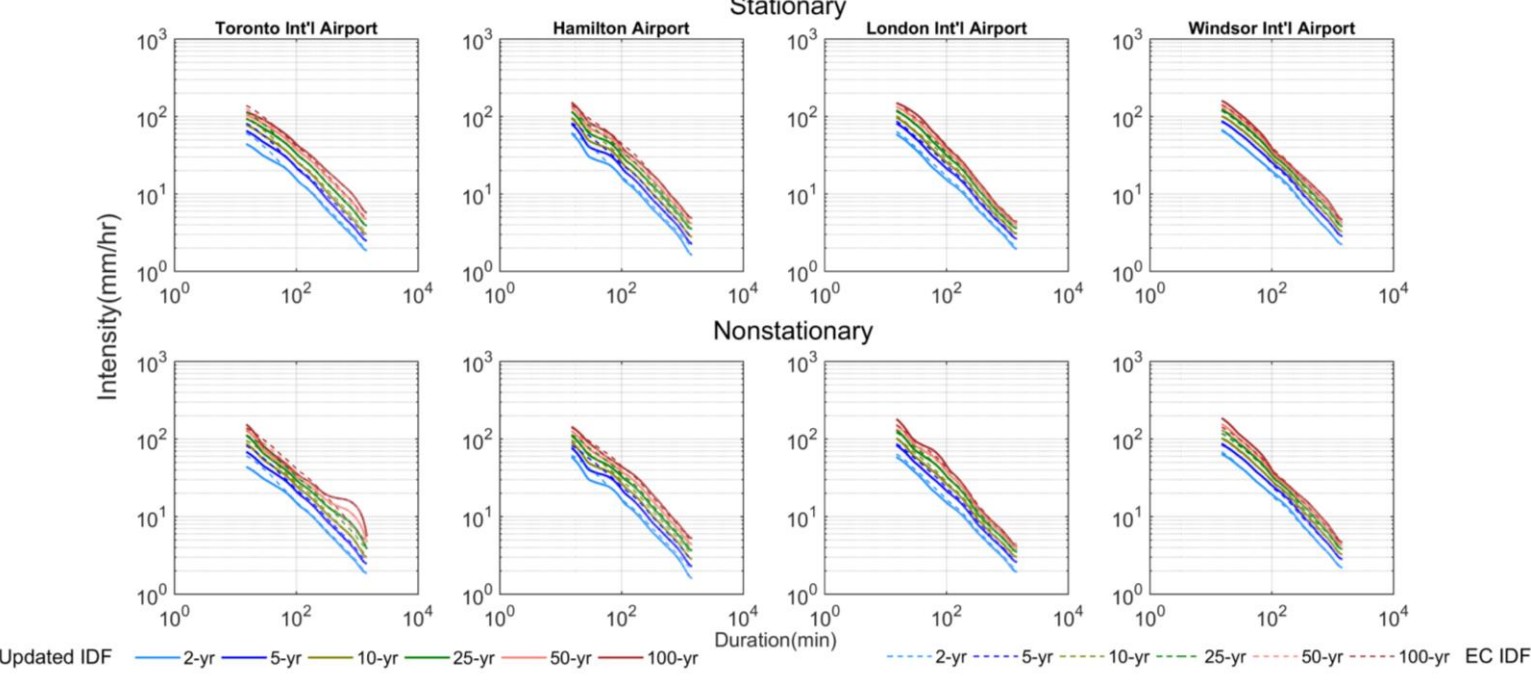

**Figure 8.** Estimated nonstationary versus EC-generated IDFs for T = 2, 5, 10, 25, 50 and 100-year return periods for the selected urbanized locations in Southern Ontario, Canada. The updated and EC IDFs are shown using solid and dotted lines respectively.