# Peer review of "Does Nonstationarity in Rainfall Requires Nonstationary Intensity-Duration-Frequency Curves?"

_Hydrology and Earth System Sciences, 2017_

## Referee Comment (RC1) · Anonymous Referee #1 · 23 Jun 2017

The manuscript presents an interesting topic, and discuss the crucial question of whether there is enough evidence of changes in hydrometric series to warrant a change in the IDF curves used for the design and maintenance of hydraulic structures.

Although the topic discussed is interesting and worthy, the paper is quite inconclusive and does not manage, in my opinion, to provide a clear point of view on the matter. The authors have definitely done a lot of work and have looked very carefully at the data, but they fail to summarise their finding in any useful way and simply provide a lot (too much maybe) of information. The presentation of the methods and results is quite unclear and it has several opaque points. The statistical methods are often

presented with some imperfections and in general the paper could greatly benefit from some proof-reading and re-organisation. In particular the authors should make more of an attempt to summarise their findings from all the non-parametric tests in a way that is more informative.

Some more specific comments

The title of the manuscript indicate that IDF curves are the main topic, although the authors limit themselves to the (hard) task of fitting different frequency curves to the each series with different duration separately. This could result in non-consistent estimates eventually. The type of studies the authors perform is laudable and would be the first step to take to assess whether new IDF curves would need to be derived.

The authors do a lot (a lot!) of tests to the data series of each duration - definitely the issue of multiple comparisons arise and it is to be expected that some tests will turn out to be significant just by randomness. I have to say it is difficult to follow the authors in all their testing, there is very little effort made to summarise the finding in any useful way and the results are simply presented/dumped as they are in the SI (except I am not sure whether the results are reliable given the authors have p-values larger than 1).

Some further remarks given in more order:

1. The beginning of Section 3.3 is very messy and should be rewritten. Distributions do not contain parameters, they are characterised by parameters. Line 25, " a value of the shape parameter equal to zero". Line 28: "In the case of a negative shape parameter, the distribution is a Weibull". Note that the Frechet is also a bounded distribution, except it has a lower bound. Overall I would write down the whole thing in a a formula, specifying the limits of the distribution for the different values of the shape.

2. Page 2 line 13. It is often the case though that IDF curves are derived not only

from at-site data but using a pooled set of stations see for Svensson and Jones (2010, doi:10.1111/j.1753-318X.2010.01079.x) for a review of methods used in several countries

3. Page 3 - line 8-9: the authors seem to imply that the Gumbel distribution is symmetric - which is not the case, as it is easy to see by plotting the pdf of a Gumbel distribution.

4. Section 3.1: I think the information of the percentage of missing values of each station/duration should be given somewhere - ideally in the main text and not in SI. I can not judge whether the MCR technique is the most appropriate one, as this is too far away from my area of expertise.

5. Page 8 - line 8: if the 5% and 95% quantile of the posterior samples are taken then a 90% credibility interval is constructed. A 95% interval is taken to be one that contains 95% of the distribution.

6. Section 3.3: it is not clear to me why the authors go through the trouble of fitting both an ML and Bayesian fit for the stationary model if they only use a Bayesian model for the non-stationary models. Just use the Bayesian methods and embrace Bayesian Inference. Also, seeing in Table SI16-S24 that the more complex non-stationary model GEVII is often selected I wander whether the authors have tried to only fit models with the scale taken as the only varying function? Lastly, why not to formally test stationary/non-stationary model is better by using a Bayesian factor or some pre-set rule on the 95% credibility interval not-containing zero?

7. Section 3.3: what do you do with the results of the Pettitt test? One could use it to build a model with a step-change rather than a continuous function of time. In general, why doing all the non-parametric test AND the parametric models? What is the use of the non-parametric tests exactly?

8. Page 8 - line 14: it is very good that the authors verify the goodness of fit by using PP, but it is unclear to me how they "select the model with fewer parameters as the best model when two models have comparable performances.". This is exactly what the AIC should do, so even if the AIC does not indicate that a simpler model should be used the authors might cull a non-stationary model out if the stationary model give a better fit in the PP plot?

9. Page 8 - line 25-26: a positive skewness is just an indication of an asymmetric/skewed distribution, it doesn't necessary indicate a change in the distribution. I mean "extreme values are more frequent in the time series" compared to what?

10. Page 9, line 29: Bayesian measures of uncertainty are normally called credibility and not confidence intervals. Also as I mentioned above - unclear if the 95% or the 90% intervals are derived.

11. Page 10/Figure 4: how are the DSI calculated for the non-stationary models? Is the last value of the parameters used to compute the quantiles? Why do you show boxplots of the posterior sample and not a 95% credibility interval? As I said I would drop the estimation using ML completely, but if you do use it, you could show confidence intervals based on the delta-method (see Coles, 2001).

12. Page 11/Figure 6: has any assessment been done on whether the stationary version of the fitted curves has a good overlap to the EC-curves? Surely if these two curves are very different, any mis-match between the non-stationary results and the EC-curves could be due to the fact that the EC curve doesn't fully fit the data of a site. This links to a comment on the statements in page 13 between line 20-25: you are saying that from the comparison of stationary to non-stationary models there seems to be no indication of a need to update DSI, but when comparing the outputs of a non-stationary model to the EC-curves (obtained assuming stationarity) then the evidence is that we should update the DSI. This points in the direction of the EC-curves being different from the at-site stationary curves.
13. Page 14: I don't understand what the last sentence of the paper means.

14. SI3: I would give the lower and upper bound of the GEV in a formula to give a simpler indication of the effect of the value of the shape parameter

15. SI3.1: why using ML in one case and Bayesian methods for another?

16. SI3.1, paragraph after equation 3.8: $p(y|\lambda, x)$ does not give information on the parameters. The formulation of the sentence seem to imply that the likelihood $p(y|\lambda, x)$ gives information on the parameters under non-stationarity, which is not the case.

17. SI4.1 - the definition in eq 4.1 for the Akaike information criterion is not correct (or better it is correct for a normal model, but not for a GEV). AIC is generally defined as AIC $= -2log(L(\omega, x)) + 2m$ . That's how the two references cited by the authors define the AIC as well. From what I understand from the explanation of the observed/expected values the authors are doing a model selection using AIC based on the quantiles, which is not made explicit in section 3.3. If that's the case, which quantiles are used?

18. Equation 5.1 and 5.2, what happens if $\zeta = 0$?

19. Table S7, 24-hours, the p-value for the pettitt test is larger than 1 - this can not be right. (see also S9 30min, S11 15min to 2hr, S14 15min to 2hr, S15 12hr)

20. Table SI16 - not sure if the red and blue are right in all stations

21. Pg 14 Supplement : the definition of return level has the word expected in the wrong place. *... often referred as return level in the literature is the expected value to be exceeded on an average once in every...* should be *... often referred as return level in the literature, is the value which is expected to be exceeded on an average once in every...* - see Coles, 2001 - end of section 3.1.3 (pg 49 in my edition)

I also find some of the Figures - and in particular their captions - could be improved

1. Figure 3 caption

   - durations higher than an hour are also shown I would say "Spatial distribution of trends, change points and non-stationarities in rainfall extremes of several durations in nine urbanized locations, Southern Ontario"
   - Drop the information on the population - it's in Figure 2 and in the text (several times)
   - Drop the information on the tests performed or at least reduce it since it's given in the text (for example drop the references)
   - Include information on the color coding in the legend.
   - If tests are performed at 5% and 10% - what is considered statistically significant? p-values < 0. 05 or p-value < 0.1?

2. Figure 4 caption: drop the list of the name of the station - it is given in the plot.

3. Figure 5 caption: add the information on the cyan shading representing the site with significant autocorrelation in the legend and drop from the legend. The second last sentence grammar is not correct.

4. Figure 7: I would include the information on solid/dotted lines in the legend

The paper has several grammar mistakes, with articles missing or appearing in the wrong place and several sentences which have non-concordant subject and verb. I list here a minuscule sample of the typos/mistakes I found

- Page 3, line 16 slowly or varying are not antonyms. Line 18-19 does should have a singular subject (not signal). Same in line 25-26.

- Page 3 line 23-24: The structure of the sentence is confusing. It is not the signatures that necessitate IDF. Maybe use "...make necessary the use..."

- Page 5: line 4-5 more repeated twice.

- Page 8: Line 27-28: the sentence is not complete

- Page 10 - line 16: less uncertainty (not lesser)

- Page 11 - line 17: More genralLY - and the sentence has a singular subject so line 19 should be is not are

- Page 12 - line 2: smaller, not lesser

- Page 12 - line 17: It? I think you need a "We"?

- Page 13 - line 6: does/is?

- Page 13 - line 12: several studies HAVE

Further inconsistencies I identified

- Page 4 Line 10: the ref to Jien and Gough is missing in the reference list and I think is not needed since it states a basic fact about the geography of Canada

- Page 9, Line 28 - $\xi$, instead of $\zeta$ used in he SI, for the shape parameter of the GEV

- Reference list: Cheng, L. and AghaKouchak, A. 2014 - just give the doi, not the ncbi link

- Supplement references: Coles and Tawn (1996) cited in text missing in the ref. Anyway, for that formula Coles, 2001 is probably enough as a citation.

- The citation to *Coles 2001, An introduction to statistical modelling of extreme values, Springer* in the supplementary material is wrong, as it has additional authors other than Coles.

---

## Referee Comment (RC2) · Anonymous Referee #2 · 25 Jul 2017

The novelty of this manuscript is applying the existing methods proposed by Cheng and AghaKouchak (2014) and Cheng et al (2014) for non-stationary IDFs to Canadian gauge rainfall. The conclusion that statistical trends in rainfall do not necessarily require the use of non-stationary IDFs is a good one. Unfortunately the impact of the results is lost in the relatively poor structure of the manuscript.

General comments:

The manuscript could do with a good proof read and rewrite. There are lots of little mistakes which makes the paper very difficult to read. I was constantly stopped in my train of reading by small errors or references to figures/tables which weren't explained.

[Figure]

The supplementary material is 66 pages and has 37 Tables. This is huge and difficult to come to terms with – I couldn't follow it all. As I don't believe a specific structure is required can I recommend the following? Group the supplementary text and figures and tables into sections. That way you will have separate sections to refer to in the main text. You can then go sequentially through the text. S1 is the infilling, S2 is the autocorrelation method and results, S3 non-stationarity test method and results, S4 GEV fitting. I may have got the headings incorrect but I hope what I mean is clear. Then with the results you can just reference a section for detailed results and focus on discussing the figures in the main text. Trying to interpret 37 tables (some split into two) – almost all which are referenced in the main text - it is like trying to read a thesis.

Moving Table S1 to the main text, and maybe removing Figure S1 altogether will make the manuscript more standalone and easier to read. This manuscript is a bit short on doing justice to some of the previous work done in this area.

Page 2 – Line 21: This is the only line discussing previous work to do with non-stationary IDFs. I think this work deserves more attention given that the focus of this manuscript is non-stationary IDFs. My recommendation is as follows:

In Page 1 – Line 23: "In a warming climate . . ." I would be a bit more careful here and expand this. I would cite Lenderink and van Meijgaard (2008) and Wasko and Sharma (2015) as papers that link temperature increases to intensifying rainfall. Most of the papers cited at the end of this sentence deal with temporal precipitation trends (and not necessarily links to temperature). It is important to make that distinction.

The reason I make the above point is the covariate used for non-stationarity is important. The authors don't raise this till the second last of their manuscript citing Mondal and Mujumdar (2015). This needs to come up in the introduction to put this manuscript novelty in context. There are more papers in this space. For example Agilan and Umamahesh (2017) and Ali and Mishra (2017) who argue for temperature to be used as a covariate (and not necessarily time). Indeed Wasko and Sharma (2017) show that

temperature is a good covariate when predicting future rainfall. Other work by Agilan and Umamahesh may also be relevant and should be discussed. Finally, I am pretty sure at least one of the Yilmaz papers suggests not much evidence (if any) for using non-stationarity so in the introduction this is not cited correctly (though I note in the discussion it is). To summarise – the literature review needs to be expanded on the above point.

Another problem I have is with the paragraph on Page 3 that starts with "secondly" – I don't think any of the research questions actually address the "secondly" point. Reading page 7 it seems you adopt the GEV and don't necessarily test this is a better fit than other distribution. This is fine – but the way this paragraph sets up the reader for something else. Either omit the "secondly" paragraph altogether or add another point to the bottom of Page 3 saying you use a GEV and the reason for doing so.

You introduce the EC data without context – so I had no idea why it was there until I got to page 11.

Top of Page 11 reads like a discussion and seems squished between the presentation of results in Figure 5 and 6. You could consider a separate discussion section and reordering of the text.

Other comments:

Page 2 – Line 16: If you are to introduce an abbreviation (TBRG) it helps to capitalise the first letter in each word before the abbreviation. This happens at several points in the text – I won't comment on the other occurrences.

Page 2 – Line 22: "The nonstationary behaviour. . ." I think I would expand this sentence to just state what places/regions the citations have studied. Reason being – in the abstract and following sentences you are referring only to Canada – so when I get to this point I am not sure if you are being Canada specific or not. Maybe this should be the start of a new paragraph and expanded a bit.

[Figure]

Page 2 – Line 26: What result? This sentence doesn't make sense – maybe some expansion of the sentences here would help.

Page 3 – Line 7: Replace "secondly" with "The second drawback of IDF curves is". You have written too much to have just the word "secondly" here. Stylistically, I don't think "first", "second" etc need to be in italics. Particularly at the bottom at Page 3 – if you are that keen on this maybe a bullet point list would be better?

Page 4 - Line 1: Remove "secondly".

Page 5 – Line 6: The reference to Table S1 doesn't belong here. I also believe Table S1 belongs in the main text.

Figure 1 – Are the record lengths for daily or sub-daily? I don't think the caption says which.

Page 5 – Line 26 – "Imputation" isn't the correct word I don't think. Infilling maybe?

Page 6 – Line 21 – Stylistically, why don't you just say "Tables S2-S4"? I do feel if you composed the supplementary material in sections you could say section S1 and be done with it.

Page 6 – Line 24 – "Figure 2 shows . . ." You are repeating a previous a sentence Section 3.2 – Is the KPSS test in Figure 2?

Page 8 – Line 4 – who else makes this assumption that only the location and scale parameter vary? I know other authors make this assumption so this assumption needs to be put in context of the other work done in this area.

Page 8 – Line 18 – So I went to the supplementary material as the text recommends and I saw four models fitted for each duration but I wasn't sure which model was which. Could this section in the main text be rewritten (maybe use some sort of list?) to say what models were fitted and clearly state their abbreviation?

Page 8 – Line 26: I disagree. Skewness of a distribution does not indicate a temporal

[Figure]

trend. This a good example of a vague sentence with a Figure in brackets (in this case Figure 3) but no mention of what I am meant to get out of looking Figure 3 in reference to this sentence. This happens throughout the text.

Figure 3 – your caption says hourly and sub-hourly. The headings in the captions go up to daily. You say you did statistical tests at 5 and 10% but don't say which final significance is presented in the plot. A legend wouldn't go astray . . .

Figure 4 – is there a particular time used for the non-stationary plots?

Page 10 – Maybe I missed this somewhere but what is the "z-statistic"? Is this the statistical test for the difference between two means?

Figure 6 – Should this have a negative scale too? Are there some sites which decrease?

References:

Lenderink, G., and E. van Meijgaard (2008), Increase in hourly precipitation extremes beyond expectations from temperature changes, Nat. Geosci., 1(8), 511–514, doi:10.1038/ngeo262.

Wasko, C., and A. Sharma (2015), Steeper temporal distribution of rain intensity at higher temperatures within Australian storms, Nat. Geosci., 8(7), 527–529, doi:10.1038/ngeo2456.

Agilan, V., and N. V Umamahesh (2017), What are the best covariates for developing non-stationary rainfall Intensity-Duration-Frequency relationship?, Adv. Water Resour., 101, 11–22, doi:10.1016/j.advwatres.2016.12.016.

Ali, H., and V. Mishra (2017), Contrasting response of rainfall extremes to increase in surface air and dewpoint temperatures at urban locations in India, Sci. Rep., 7(1), 1228, doi:10.1038/s41598-017-01306-1.

Wasko, C., and A. Sharma (2017), Continuous rainfall generation for a warmer

climate using observed temperature sensitivities, J. Hydrol., 544, 575–590, doi:10.1016/j.jhydrol.2016.12.002.

---

## Author Comment (AC1) · 16 Aug 2017

**Responses to Reviewer #1 on "Does Nonstationarity in Rainfall Requires Nonstationary Intensity-Duration-Frequency Curves? By Poulomi Ganguli and Paulin Coulibaly**

We thank Referee #1 for reviewing our manuscript and providing constructive feedback, which improves the quality of the manuscript. Our responses are embedded within the comments (in BLACK) in BLUE.

The manuscript presents an interesting topic, and discuss the crucial question of whether there is enough evidence of changes in hydrometric series to warrant a change in the IDF curves used for the design and maintenance of hydraulic structures. Although the topic discussed is interesting and worthy, the paper is quite inconclusive and does not manage, in my opinion, to provide a clear point of view on the matter. The authors have definitely done a lot of work and have looked very carefully at the data, but they fail to summarize their finding in any useful way and simply provide a lot (too much maybe) of information. The presentation of the methods and results is quite unclear and it has several opaque points**.** The statistical methods are often presented with some imperfections and in general the paper could greatly benefit from some proof-reading and re-organisation. In particular the authors should make more of an attempt to summarise their findings from all the non-parametric tests in a way that is more informative.

**Response:** Thanks for the feedback. The reviewer comments are well appreciated. In our case, a series of statistical tests are necessary to assess nonstationarity in design rainfall, as echoed in earlier literature (Sadri et al., 2016; Yilmaz et al., 2014, 2017). A single statistical test may not be reliable enough to detect signatures of nonstationarity in hydrometeorological time series. Further, we note that multiple tests allow a more rigorous assessment of overall trend in the time series since certain tests are complimentary to each other. Therefore, we explored various statistical tests, starting from testing auto-correlation, presence of monotonic (using trend tests) or abrupt change (using single point change detection algorithm) at different statistical significance levels in practice. Further, we have presented a flowchart of complete methodology in Figure 2 to comprehend the overall analysis. Now coming to statistical methods, we have significantly revised the manuscript to correct any miss perfections as pointed by the reviewer. While we are highly appreciative of the suggestions and comments by the reviewers, we do have one minor point to make which may come across as a slight disagreement with one set of comments. We sense a sentiment shared in one of the comments that our presentation of methods and results are quite unclear. We do not agree with this sentiment even though we agree that the various nuances were not clearly explained in the previous version of the manuscript. Since the focus of the work is insight driven, we have discussed methodologies thoroughly in Supplements to avoid distraction of audience by over-emphasizing the methodologies.

However, we have attempted to improve the presentation of methods and re-organized our manuscript in light of the reviewer's comments. As suggested we have made the following changes in the revised manuscript:

- We have expanded Section 3.2 in Methods to include rationale for the inclusion of multiple tests for detecting nonstationarity. We argue that some of the tests are complimentary to each other. Further, multiple tests allows a robust assessment of overall trend, shifts and nonstationarity in the time series as suggested in the literature (Sadri et al., 2016; Yilmaz et al., 2014, 2017).
- We have reorganized Section 3.3 to include mathematical formulations of GEV distribution and associated time varying covariates to model nonstationary GEV parameters.
- We have re-written the Methodology section and re-organized the Supplements into different sections to present it in a more coherent and clearer way to the readers.
- We have summarized the results of trend detection tests in detail in Page 12, line $17 - 29$.
- We have included Bayes-factor criterion in addition to AIC statistics for small sample to evaluate fit of the nonstationary model.
- We have restricted our analysis to Bayesian fit for stationary and nonstationary model.
- We have recalculated 95% credible intervals for all sites from 0.025 and 0.975 quantiles of the simulated posterior samples.

The title of the manuscript indicate that IDF curves are the main topic, although the authors limit themselves to the (hard) task of fitting different frequency curves to the each series with different duration separately. This could result in non-consistent estimates eventually. The type of studies the authors perform is laudable and would be the first step to take to assess whether new IDF curves would need to be derived.

**Response:** Here we slightly disagree with the reviewer. First, we fitted both stationary and nonstationary frequency curves corresponding standard durations, commonly used in practice for infrastructure in design. We also test the hypothesis whether we need nonstationary frequency curves for the moderately and densely populated urbanized locations across Southern Ontario. We discussed motivation of our study in detail and extended literature review in the revision. Next, we compared the design storm estimates using simple z-statistics considering range of uncertainty as assessed by 95% credible interval to find out whether statistically significant differences exist between nonstationary versus stationary method. Finally, we presented updated IDF curves for all nine locations across Southern Ontario, which is of interest to stakeholders' of the region. We further compared updated versus EC-generated IDFs considering both nonstationary and stationary (Figures 6 and 7) conditions.

The authors do a lot (a lot!) of tests to the data series of each duration - definitely the issue of multiple comparisons arise and it is to be expected that some tests will turn out to be significant just by randomness.

**Response:** We appreciate the reviewer's point. However, we would stress that multiple tests are needed to detect presence of monotonic trends or abrupt shifts, and nonstationarity in the time series since a selected or cherry-picked number of tests may not be sufficient to detect plausible changes and nonstationarity in the time series. Multiple tests were also performed in earlier studies (Sadri et al., 2016; Yilmaz et al., 2014, 2017) to detect temporal changes in the time series. For example, we employ both Mann-Whitney and Pettitt method to find abrupt shift in mean in the time series, whereas Mann-Kendall test was employed to detect monotonic trend in the time series.

Previous studies (Xie et al., 2014; Yue and Wang, 2002) have found that the rank-based nonparametric Mann-Whitney test is not really distribution free and the power of test is often affected by the properties of sampled data. In practice, when real change point is unknown, often Mann-Whitney test in general does not work well, and the Pettitt method can yields plausible change point location along with its statistical significance. However, significance of the Pettitt test can be obtained using an approximated limiting distribution. As shown earlier, the p-value associated with the test statistics is evaluated following an approximate estimate (Xie et al., 2014). Further, it is also important to note that presence of nonstationarity may not be evaluated merely on the basis of trends or abrupt shifts in the time series, even if the increasing or decreasing trends are statistically significant (Yilmaz et al., 2014). Therefore, we also employed three statistical tests, namely Augmented Dickey-Fuller (ADF), Kwiatkowski–Phillips–Schmidt–Shin (KPSS) and Priestley Subbarao (PSR) test to further investigate nonstationarity in the time series. Both ADF and KPSS tests are based on autoregressive nature of time series. However, Yilmaz et al. (2014) did not observe presence of any significant nonstationarity in short-duration extreme rainfall time series in the city of Melbourne even after employing these tests. Therefore as an alternative, we employed frequency-based PSR test, which is able to capture nonlinear dynamical nature of hydrological system than the former two tests (Ali and Mishra, 2017; Hamed and Rao, 1999). We have incorporated these points in the revised version of the manuscript in appropriate places (Page 8, lines 20-24; Page 9, lines 2 - 6).

I have to say it is difficult to follow the authors in all their testing, there is very little effort made to summarise the finding in any useful way and the results are simply presented/dumped as they are in the SI.

**Response:** We agreed. In the revised manuscript we provided a more detailed description of the results:

- In page 11, lines 21-26, we provided results of skewness and kurtosis in Annual Maxima (AM) time series. We move results of skewness and kurtosis analysis in the form of Tables (Tables 2 and 3) in main manuscript. We have added following sentences:

  "The skewness is a measure of the asymmetry in the AMP distribution. Positive values of skewness indicate that data are skewed to the right. The skewness of sub-hourly precipitation extremes varies between 0.22 and 4.45, with highest being 30-min AMP record at Hamilton and least being at Oshawa respectively (Table 2). Likewise, for hourly extremes, the skewness ranges between 0.54 and 2.54, with least being 1-hour AMP at Oshawa and highest is 1-hour AMP at Hamilton respectively (Table 3)."

- In page 12, lines 17 – 29, we summarized results of nonstationary trend detection tests. We have added following sentences in the revised manuscript:

  "We find statistically significant monotonic increase and abrupt step changes, both in mean and variance in Oshawa and Trenton respectively (Table S6 and S10), whereas London show (significant) decrease (Table S9) from duration of 6-hour and more. Windsor, Kingston and Stratford show (significant) step changes as confirmed by Mann-Whitney and Mood Tests (Tables S7, S8 and S11). On the other hand, Toronto, Hamilton and Fergus Shand Dam (Tables S4, 4.1; S5, 5.1; S12) do not exhibit any statistically significant gradual or abrupt changes in the AMP time series. The ADF tests show presence of nonstationarity in all durations across the sites. To further validate results of ADF test, KPSS and PSR tests are employed. The KPSS test detects presence of nonstationarity at 3 out of 9 sites for 24-hour rainfall extreme at 5% significance level, whereas the results of PSR test indicate nonstationarity across 5 sites in 24-hour rainfall extremes. While KPSS test alone could not detect presence of nonstationarity in any of the extreme series in Oshawa and Stratford respectively, the results of PSR test did not indicate nonstationarity in any of the short-duration rainfall extreme in Windsor. Both of these tests taken together detect presence of nonstationarity in rainfall extremes across 6 out of 9 sites".

- We have incorporated results of the nonstationary versus stationary model fit of selected airport sites, such as, Toronto, Hamilton, Windsor and London in Tables 4 – 7 in the main manuscript and explained the results in page 13, lines 9 – 14.

- We have revised the result section to include more thorough explanation of each of the findings.

I am not sure whether the results are reliable given the authors have p-values larger than 1.

**Response:** Here we briefly explain computation procedure of Pettitt Test (Xie et al., 2014), which we have appended in Supplements (SI 2).

When a sequence of random variables is divided into two segments represented by $x_1, ..., x_{t_0}$ and $x_{t_0+1}, x_{t_0+2}, ..., x_{t_0}$, if each segment has distribution functions, $F_1(x)$ and $F_2(x)$, where $F_1(x) \neq F_2(x)$, then change point is identified at $t_0$. Thus the null hypothesis of the test is "no change", $H_0 : \tau = T$ against the alternative of "change" $H_1 : 1 \leq \tau < T$. The test is based on following statistic (Serinaldi and Kilsby, 2016; Xie et al., 2014)

$$K_T = \max_{1 \leq t \leq T} |U_{t,T}|, \text{ where } U_{t,T} = \sum_{i=1}^{t} \sum_{j=i+1}^{T} \text{sgn}(X_i - X_j)$$

Where $\text{sgn}(x) = 1$, if $x > 0$, 0 if $x = 0$ and -1 if $x < 0$. The p-value associated with $K_T$ is approximately evaluated as (Xie et al., 2014), $p = 2 \exp\left(\dfrac{-6K_T^2}{T^2 + T^3}\right)$. Given a certain significance level $\alpha$, if $p < \alpha$, we reject the null hypothesis and conclude that $x_\tau$ is a significant change point at level $\alpha$. Since the associated *p-value* is computed following an approximate estimate of *p-value*, in few cases it exceeds the value 1, which we sense is due to analytical intractability of the estimate. In that case, we have kept the table value blank simply putting a hyphen, and added a footnote indicating the calculation of *p-value* is analytically intractable in those cases.

**Response to Further Remarks by Reviewer 1**

**Comment 1** The beginning of Section 3.3 is very messy and should be rewritten. Distributions do not contain parameters, they are characterised by parameters. Line 25, " a value of the shape parameter equal to zero". Line 28: "In the case of a negative shape parameter, the distribution is a Weibull". Note that the Frechet is also a bounded distribution, except it has a lower bound. Overall I would write down the whole thing in a formula, specifying the limits of the distribution for the different values of the shape.

**Response:** Agreed. We have added following sentences in the revision:

"The GEV distribution is characterized by three parameters, the location, the scale and the shape of the distribution, which describes the center of the distribution, the deviation around the mean

and the shape or the tail of the distribution (Katz et al., 2002; Katz and Brown, 1992). The cumulative distribution function of stationary (time invariant) GEV model is given by (Coles et al., 2001):

$$G(z) = \exp\left\{-\left[1+\zeta\left(\frac{z-\mu}{\sigma}\right)\right]_+^{-1/\zeta}\right\} \qquad \sigma > 0, -\infty < \mu, \zeta < \infty \qquad (3.1)$$

Where, $y_+ = \max\{y, 0\}$, and

$z \in \left[(\mu-\sigma)/\zeta, +\infty\right)$ when $\zeta > 0$; $z \in \left(-\infty, (\mu-\sigma)/\zeta\right]$ when $\zeta < 0$; and $z \in (-\infty, +\infty)$ when $\zeta = 0$

$\mu$ is a location parameter, $\sigma$ is a scale parameter and $\zeta$ is a shape parameter determining the heaviness of the tail. The shape parameter $\zeta$, determines the higher moments of the density function and also the skew in the probability mass. The '+' sign indicates positive part of the argument. The Eq. (3.1) encompasses three types of DFs based on the sign of the shape parameter, $\zeta$ : (i) the Fréchet, with a finite lower bound of $(\mu-\sigma)/\zeta$ and an unbounded, heavy upper tail, ($\zeta > 0$), (ii) the Weibull, unbounded below and with a finite upper bound of $(\mu-\sigma)/\zeta$, ($\zeta < 0$) and (iii) the Gumbel, unbounded below and above with a light upper tail $\zeta = 0$, formally obtained by taking limit as $\zeta \to 0$. The Gumbel distribution is described by an unbounded light tailed distribution and the tail decreases rapidly following an exponential decay. The Fréchet distribution is a heavy-tailed distribution, and the tail drops relatively slowly following a polynomial decay (Towler et al., 2010). On the other hand, the Weibull distribution is a bounded distribution".

**Comment 2** Page 2 line 13. It is often the case though that IDF curves are derived not only from at-site data but using a pooled set of stations see for Svensson and Jones (2010, doi:10.1111/j.1753-318X.2010.01079.x) for a review of methods used in several countries.

**Response:** Agreed. The approach can be implemented locally (at Site; or SFA) or regionally (RFA or pooled). The regional frequency analysis is used when available record length are short or at locations where no observed data are available (Castellarin et al., 2012; Komi et al., 2016). However, various RFA estimation methods have certain drawbacks, such as Index flood method is sensitive to the homogeneity assumption and formation of regions; in Bayesian method of regionalization, the prior distributions of parameters are often not precise enough and do not add precision to the estimates; in Hierarchical approach, the method may produce abrupt changes in

the parameters from one site to another. Komi et al. (2016) summarizes the limitations and advantages of some of the widely used RFA techniques. In our case the available records across all sites ranges between 47 and 66 years, which are more than the climatology (often over time periods of 30-years) of a region. Hence, we employ SFA method in our study. The rationale of incorporating at-site frequency method to derive IDF curves in the present study is discussed briefly in page 3, lines 17 – 25. This also allows a consistent comparison with the EC-IDFs that have been used in practice in the study area.

**Comment 3** Page 3 - line 8-9: the authors seem to imply that the Gumbel distribution is symmetric - which is not the case, as it is easy to see by plotting the pdf of a Gumbel distribution.

**Response:** We agree. This was a mistake. We revise the sentence as follows:

EV1 distribution has certain limitations, such that it is a non-heavy tailed distribution and characterized by constant skewness and kurtosis coefficients.

**Comment 4** Section 3.1: I think the information of the percentage of missing values of each station/duration should be given somewhere - ideally in the main text and not in SI. I can not judge whether the MCR technique is the most appropriate one, as this is too far away from my area of expertise.

**Response:** We agree. We have moved Table S1 from Supplement to main manuscript as Table 1. We have also added an extra column in Table 1 indicating information of missing years and durations at each station.

**Comment 5** Page 8 - line 8: if the 5% and 95% quantile of the posterior samples are taken then a 90% credibility interval is constructed. A 95% interval is taken to be one that contains 95% of the distribution.

**Response:** Agreed! As suggested we have re-analyzed our data to incorporate 2.5% and 97.5% quantiles of the posterior sample to construct a 95% credible interval.

**Comment 6** Section 3.3: it is not clear to me why the authors go through the trouble of fitting both an ML and Bayesian fit for the stationary model if they only use a Bayesian model for the non-stationary models. Just use the Bayesian methods and embrace Bayesian Inference.

**Response:** We appreciate the reviewer's comment. As suggested, we have presented the results only using Bayesian model and exclude ML method.

Also, seeing in Table SI16-S24 that the more complex non-stationary model GEVII is often selected I wander whether the authors have tried to only fit models with the scale taken as the only varying function?

**Response:** We have revised our results in light of the above comments. However, from revised set of results we noted that in a few cases GEV II model (nonstationary in location and scale parameter), performed better than GEV I model (nonstationary in location only). The above results are not uncommon given the highly nonstationary nature of precipitation extremes as observed from the Figure 3. Similar findings were also noted by (Gu et al., 2017) in a flood frequency analysis of Pearl River basin in China, where the author have analyzed 28 stream gauge locations. The results of their analysis suggested in 5 out of 28 sites GEVII performed better as compared to the stationary and GEV I models.

Lastly, why not to formally test stationary/non-stationary model is better by using a Bayesian factor or some pre-set rule on the 95% credibility interval not-containing zero?

**Response:** Agreed! We have incorporated Bayes factor, AIC statistics for small sample and probability-probability (*P-P*) plot to evaluate model fit.

**Comment 7** Section 3.3: what do you do with the results of the Pettitt test? One could use it to build a model with a step-change rather than a continuous function of time. In general, why doing all the non-parametric test AND the parametric models? What is the use of the non-parametric tests exactly?

**Response:** This is indeed a good point raised by the reviewer. Here, we used three different tests, Pettitt, Mann-Whitney and Mood tests to identify abrupt step changes in the time series, which is different from monotonic or gradual trends in the time series. We have implemented a series of statistical tests since a single statistical test may not be able to capture full ranges of nonstationarity in highly nonlinear dynamical system, such as short-duration extreme precipitation. As we discussed earlier, the rank-based nonparametric Mann-Whitney test is not really a distribution free and the power of the test is often affected by the properties of sampled data. In practice, when real change point is unknown, often Mann-Whitney test in general does not work well and the Pettitt method can yield plausible change point location along with its statistical significance. However, the significance of the Pettitt test can be obtained using an approximated limiting distribution. Therefore, above tests were needed in the current setting.

Further, we applied nonparametric tests due to their robustness to non-normality, which usually appears in the hydroclimatic time series. Further, in order to reduce the number of underlying assumption required for testing a hypothesis, such as presence of specific kind of trend or change

point in the data set, nonparametric tests were employed. We discussed each of these issues in the revised manuscript.

**Comment 8.** Page 8 - line 14: it is very good that the authors verify the goodness of fit by using PP, but it is unclear to me how they "select the model with fewer parameters as the best model when two models have comparable performances.". This is exactly what the AIC should do, so even if the AIC does not indicate that a simpler model should be used the authors might cull a non-stationary model out if the stationary model give a better fit in the PP plot?

**Response:** We have reanalyzed the data and new results are different from the previous ones.

**Comment 9.** Page 8 - line 25-26: a positive skewness is just an indication of an asymmetric/skewed distribution, it doesn't necessary indicate a change in the distribution. I mean "extreme values are more frequent in the time series" compared to what?

**Response:** We have revised this sentence in page 11 (line 22-23) as follows:

Positive values of skewness indicate that data are skewed to the right.

**Comment 10.** Page 9, line 29: Bayesian measures of uncertainty are normally called credibility and not confidence intervals. Also as I mentioned above - unclear if the 95% or the 90% intervals are derived.

**Response:** We appreciate reviewer's feedback. As suggested we have replaced the word with credibility interval wherever it is appropriate. We have constructed 95% credibility intervals from the 2.5$^{th}$ and 97.5$^{th}$ percentiles of the simulated posterior samples.

**Comment 11.** Page 10/Figure 4: how are the DSI calculated for the non-stationary models? Is the last value of the parameters used to compute the quantiles? Why do you show boxplots of the posterior sample and not a 95% credibility interval? As I said I would drop the estimation using ML completely, but if you do use it, you could show confidence intervals based on the delta-method (see Coles, 2001).

**Response:** We estimated parameters using Bayesian inference (BI) coupled with Differential Evaluation Markov Chain (DE-MC) simulation as in (Cheng and AghaKouchak, 2014; Cheng et al., 2014). DE-MC is an adaptive Monte Carlo Markov Chain (MCMC) algorithm (Ter Braak and Vrugt, 2008; Ter Braak, 2006), in which multiple chains (here, we fix chain length '*n*' as 5) are run in parallel. The resulting MC simulations are then run to an equilibrium (often referred to as the *burn-in* period). It is a standard practice to discard the initial iterations of simulated samples since they are strongly influenced by starting values and do not provide usable

information of the target distribution. Here we run DE-MC simulations for 3000 iterations and kept the 2001-3000[th] iterations of each chain. The convergence of MC simulation is checked by the "potential scale reduction factor $(\widehat{R})$" as in (Gelman et al., 2011), which suggests the value of $\widehat{R}$ should remain below the threshold value of 1.1. The post burn-in random draws from posterior distribution is then used to construct predictive distributions. For annual maxima time series of each duration, the mean and associated 95% credibility intervals of parameters $(\mu(t), \sigma(t))$ are derived by computing 50[th] (the median), 2.5[th] and 97.5[th] (bounds) percentiles of post *burn-in* random draw (for example, 50[th] percentile of $\mu(t_1), \ldots, \mu(t_{100})$). The derived model parameters are then used to compute corresponding design rainfall quantiles at *T*-year return period and corresponding credibility interval. We calculated median value of design storm by computing 50[th] percentiles of the post-burn in simulated posterior quantiles for the nonstationary model. We have constructed 95% credibility intervals from the 2.5[th] and 97.5[th] percentiles of the posterior samples. In the manuscript, the boxplots are shown for 95% credibility interval and not with posterior samples. To avoid further ambiguity we have revised corresponding figure caption (Figure 4) as, "DSI estimates of median (horizontal line within the box plot) and 95% credible intervals for 100-year return periods of stationary versus nonstationary models across nine sites (a - i). The boxplots indicate the uncertainty in estimated DSI using Bayesian inference". As suggested we have dropped ML method completely in the revised manuscript.

**Comment 12.** Page 11/Figure 6: has any assessment been done on whether the stationary version of the fitted curves has a good overlap to the EC-curves? Surely if these two curves are very different, any mis-match between the non-stationary results and the EC-curves could be due to the fact that the EC curve doesn't fully fit the data of a site. This links to a comment on the statements in page 13 between line 20-25: you are saying that from the comparison of stationary to non-stationary models there seems to be no indication of a need to update DSI, but when comparing the outputs of a non-stationary model to the EC-curves (obtained assuming stationarity) then the evidence is that we should update the DSI. This points in the direction of the EC-curves being different from the at-site stationary curves.

**Response:** As suggested we have compared stationary version of the fitted curve with EC curves. Associated results are presented in Figures 8 and S15. We discuss following results in the revision:

"In order to distinguish between stationary and nonstationary method of analysis, we also present updated IDF assuming stationary condition relative to EC IDF in the same plot (in top panel). The comparisons of remaining sites are presented in Figure S15. Thus we made the first attempt to compare the results of updated versus EC-generated IDFs considering both nonstationary and

stationary conditions, which are part of contemporary Design Standards and widely used by the stakeholders and practitioners. Overall, the updated IDFs closely follow the pattern of trends analogous to EC-generated IDFs, except for the 100-year return period. The difference being more pronounced considering nonstationary condition, especially at Toronto International Airport (Figure 8), Oshawa WPCP and Stratford WWTP (Figure S15). At longer durations and higher return periods, stations in metropolitan areas (such as Toronto International Airport, Hamilton Airport, Oshawa WPCP and Windsor Airport) show large differences in DSIs, whereas moderately populated locations such as, Kingston P. station and Fergus Shand dam show relatively smaller changes. Considering, nonstationary condition, the maximum increase in Furgas Shand dam is noted as 18.7% for the 2-hour storm duration and 100-year return period, whereas an increase of around 44.5% is shown for 12-hour storm duration at Toronto Airport".

**Comment 13.** Page 14: I don't understand what the last sentence of the paper means.

**Response:** We have revised the sentence as follows:

"Given that these findings are for the current period (e.g. historical extreme rainfall time series), we recommend a careful extrapolation of the findings with regards to future climate projections, in which frequency and magnitude of extreme rainfall are expected to intensify (Mailhot et al., 2012; Deng et al., 2016; Fischer and Knutti, 2016; Prein et al., 2016; Pfahl et al., 2017)". Further work should consider nonstationary methods for deriving future IDFs in Southern Ontario.

**Comment 14.** SI3: I would give the lower and upper bound of the GEV in a formula to give a simpler indication of the effect of the value of the shape parameter.

**Response:** Agreed, we add following expressions to indicate effect of shape parameter in GEV distribution:

$$G(z) = \exp\left\{-\left[1+\zeta\left(\frac{z-\mu}{\sigma}\right)\right]_+^{-1/\zeta}\right\} \qquad \sigma > 0, -\infty < \mu, \zeta < \infty \tag{3.1}$$

Where, $y_+ = \max\{y, 0\}$, and

$z \in \left[(\mu - \sigma)/\zeta, +\infty\right)$ when $\zeta > 0$; $z \in \left(-\infty, (\mu - \sigma)/\zeta\right]$ when $\zeta < 0$; and $z \in (-\infty, +\infty)$ when $\zeta = 0$

$\mu$ is a location parameter, $\sigma$ is a scale parameter and $\zeta$ is a shape parameter determining the heaviness of the tail. The shape parameter $\zeta$, determines the higher moments of the density function and also the skew in the probability mass. The '+' sign indicates positive part of the argument. The Eq. (3.1) encompasses three types of DFs based on the sign of the shape parameter, $\zeta$: (i) the Fréchet, with a finite lower bound of $(\mu - \sigma)/\zeta$ and an unbounded, heavy upper tail, ($\zeta > 0$), (ii) the Weibull, unbounded below and with a finite upper bound of $(\mu - \sigma)/\zeta$, ($\zeta < 0$) and (iii) the Gumbel, unbounded below and above with a light upper tail $\zeta = 0$, formally obtained by taking limit as $\zeta \to 0$.

**Comment 15.** SI3.1: why using ML in one case and Bayesian methods for another?
**Response:** Agreed. As suggested we have excluded the results of ML estimate.

**Comment 16.** SI3.1, paragraph after equation 3.8: p(y|λ, x) does not give information on the parameters. The formulation of the sentence seem to imply that the likelihood p(y|λ, x) gives information on the parameters under non-stationarity, which is not the case.
**Response:** Agreed. To avoid any ambiguity, we have revised the sentence as:
The posterior distributions, $p(\omega|y)$ and $p\left(y| \lambda, x\right)$ indicate likelihood functions, which infer parameters $\omega = \{\mu, \sigma, \zeta\}$ considering stationarity, and $\lambda = \{\mu_1, \mu_0, \sigma_1, \sigma_0, \zeta\}$ assuming nonstationarity conditions, respectively.

**Comment 17.** SI4.1 - the definition in eq 4.1 for the Akaike information criterion is not correct (or better it is correct for a normal model, but not for a GEV). AIC is generally defined as AIC = −2log(L(ω, x)) + 2m . That's how the two references cited by the authors define the AIC as well. From what I understand from the explanation of the observed/expected values the authors are doing a model selection using AIC based on the quantiles, which is not made explicit in section 3.3. If that's the case, which quantiles are used?

**Response:** Here we cannot concur with the reviewer. We also point to the reviewer that we have used a least square version of Akaike Information Criterion (*AIC*), which is calculated as the largest deviation between the observed (empirical in this case, obtained from rank-based plotting

position formula) and modelled cumulative distribution. This form of *AIC* is widely used in hydrology in general and multivariate statistics in particular (Dawson et al., 2007; Deepthi Rajsekhar et al., 2015; Ganguli and Reddy, 2012; Hu, 2007; Janga Reddy and Ganguli, 2012; Karmakar and Simonovic, 2007, 2009). Further, we point that this form does not correspond to a normal model. For calculation of *AIC* statistics, we consider median of the DE-MC sampled parameters, which can be considered as an average or expected value of risk in the historical observation. We have added this in detail in section 3.3 as suggested by the reviewer.

**Comment 18**. Equation 5.1 and 5.2, what happens if $\zeta = 0$?

**Response:** When $\zeta \rightarrow 0$, the GEV distribution reduces to Gumbel distribution (or Extreme Value Type I). In that case, the return period is obtained by calculating frequency factor. We add following sentences SI 4, page 40 in the revised version of the manuscript:

"When $\zeta \rightarrow 0$, the GEV distribution reduces to Gumbel distribution (or Extreme Value Type I). It should be noted that Gumbel Extreme value distribution has been commonly usto estimate design storm by Environment Canada (CSA, 2010). The Gumbel probability distribution has following form (Wang et al., 2015)

$$q_p = \mu + K_p \sigma$$

Where $K_p$ denotes frequency factor depending on the return period *T*, which is obtained using following relationship (Wang et al., 2015)

$$K_p = \frac{-\sqrt{6}}{\pi} \left[ 0.5772 + \ln\left( \ln\left( \frac{T}{T-1} \right) \right) \right]$$

Environment Canada uses this method to estimate rainfall frequency at a given duration and obtain nationwide IDF curves".

**Comment 19** Table S7, 24-hours, the p-value for the pettitt test is larger than 1 - this cannot be right. (see also S9 30min, S11 15min to 2hr, S14 15min to 2hr, S15 12hr)

**Response:** Agreed. As explained before, the significance of the Pettitt test can be obtained using an approximated limiting distribution, the p-value of certain durations could not be computed accurately due to analytical intractability. We have kept those places as blank (-) in the revised manuscript. We have added a footnote at the end of Table S4 explaining this point.

**Comment 20** Table SI16 - not sure if the red and blue are right in all stations.

**Response:** We have revised our analysis and revised results are different from earlier.

**Comment 21** Pg 14 Supplement : the definition of return level has the word expected in the wrong place. ... often referred as return level in the literature is the expected value to be exceeded on an average once in every... should be ... often referred as return level in the literature, is the value which is expected to be exceeded on an average once in every... - see Coles, 2001 - end of section 3.1.3 (pg 49 in my edition).

**Response:** Agreed. We have revised the definition in current version as suggested.

**Comment 22** I also find some of the Figures - and in particular their captions - could be improved.

**Response:** Agreed. We have revised captions of the figures wherever appropriate to enhance clarity. By doing so, we have also incorporated changes as suggested by the reviewer.

**Comment 22.1** Figure 3 caption
- Durations higher than an hour are also shown I would say "Spatial distribution of trends, change points and non-stationarities in rainfall extremes of several durations in nine urbanized locations, Southern Ontario"
- Drop the information on the population - it's in Figure 2 and in the text (several times)
- Drop the information on the tests performed or at least reduce it since it's given in the text (for example drop the references)
- Include information on the color coding in the legend.
- If tests are performed at 5% and 10% - what is considered statistically significant? p-values < 0. 05 or p-value < 0.1?

**Response:** Agreed and incorporated in the revision. Further, p-values < 0.1 is considered to be statistically significant. The same has been incorporated in the revision.

**Comment 22.2** Figure 4 caption: drop the list of the name of the station - it is given in the plot.

**Response:** Agreed and incorporated in the revision. Also we have revised the figure caption in light of comment no. 11.

**Comment 22.3** Figure 5 caption: add the information on the cyan shading representing the site with significant autocorrelation in the legend and drop from the legend. The second last sentence grammar is not correct.

**Response:** Agreed and incorporated in the revision. We have revised the grammar of the second last sentence.

**Comment 22.4** Figure 7: I would include the information on solid/dotted lines in the legend.

**Response:** Agreed and incorporated in the revision.

**Comment 23** The paper has several grammar mistakes, with articles missing or appearing in the wrong place and several sentences which have non-concordant subject and verb. I list here a minuscule sample of the typos/mistakes I found

**Response:** We have thoroughly checked the manuscript, corrected all typos. We have revised the manuscript in places as they were suggested.

**Comment 23.1** Page 3, line 16 slowly or varying are not antonyms. Line 18-19 does should have a singular subject (not signal). Same in line 25-26.

**Response:** Agreed. We have revised this to gradual or monotonic changes. We have revised the sentence in line 18-19. We have revised the grammar in line 25-26.

**Comment 23.2** Page 3 line 23-24: The structure of the sentence is confusing. It is not the signatures that necessitate IDF. Maybe use "...make necessary the use..."

**Response:** Agreed and incorporated in the revision.

**Comment 23.3** Page 5: line 4-5 more repeated twice.

**Response:** Agreed and we have revised the sentence as suggested.

**Comment 23.4** Page 8: Line 27-28: the sentence is not complete.

**Response:** We apologized for this. We have corrected all incomplete sentences including this one in the revision.

**Comment 23.5** Page 10 - line 16: less uncertainty (not lesser).

**Response:** Agreed and incorporated in the revision.

**Comment 23.6** Page 11 - line 17: More genralLY - and the sentence has a singular subject so line 19 should be is not are.

**Response:** Agreed and incorporated in the revision.

**Comment 23.7** Page 12 - line 2: smaller, not lesser.

**Response:** Agreed and incorporated in the revision.

**Comment 23.8** Page 12 - line 17: It? I think you need a "We"?

**Response:** Agreed and incorporated in the revision.

**Comment 23.9** Page 13 - line 6: does/is?

**Response:** Agreed and incorporated in the revision.

**Comment 23.10** Page 13 - line 12: several studies HAVE.

**Response:** Agreed and incorporated in the revision.

**Comment 24.** Further inconsistencies I identified:

- **Comment 24.1** Page 4 Line 10: the ref to Jien and Gough is missing in the reference list and I think is not needed since it states a basic fact about the geography of Canada.
  **Response:** Agreed and the citation is excluded from the revised version.

- **Comment 24.2** Page 9, Line 28 - $\xi$, instead of $\zeta$ used in the SI, for the shape parameter of the GEV.
  **Response:** Agreed and incorporated in the revised version of the manuscript.

- **Comment 24.3** Reference list: Cheng, L. and AghaKouchak, A. 2014 - just give the doi, not the ncbi link.
  **Response:** Agreed and incorporated in the revision.

- **Comment 24.3** Supplement references: Coles and Tawn (1996) cited in text missing in the ref. Anyway, for that formula Coles, 2001 is probably enough as a citation.
  **Response:** The citation Coles and Tawn (1996) is included in the revised version.

- **Comment 24.4** The citation to Coles 2001, An introduction to statistical modelling of extreme values, Springer in the supplementary material is wrong, as it has additional authors other than Coles.
  **Response:** Agreed and incorporated in the revision.

**References**

Ali, H. and Mishra, V.: Contrasting response of rainfall extremes to increase in surface air and dewpoint temperatures at urban locations in India, Sci. Rep., 7(1), 1228, doi:10.1038/s41598-017-01306-1, 2017.

Castellarin, A., Kohnová, S., Gaál, L., Fleig, A., Salinas, J. L., Toumazis, A., Kjeldsen, T. R. and Macdonald, N.: Review of applied-statistical methods for flood-frequency analysis in Europe, Available from: http://nora.nerc.ac.uk/19286/, 2012.

Cheng, L. and AghaKouchak, A.: Nonstationary precipitation intensity-duration-frequency curves for infrastructure design in a changing climate, Sci. Rep., 4, doi: 10.1038/srep07093, 2014.

Cheng, L., AghaKouchak, A., Gilleland, E. and Katz, R. W.: Non-stationary extreme value analysis in a changing climate, Clim. Change, 127(2), 353–369, 2014.

Coles, S. G. and Tawn, J. A.: A Bayesian Analysis of Extreme Rainfall Data, J. R. Stat. Soc. Ser. C Appl. Stat., 45(4), 463–478, doi:10.2307/2986068, 1996.

Coles, S.: An introduction to statistical modeling of extreme values, Springer, 2001.

CSA (Canadian Standards Association): Technical Guide – Development, Interpretation and Use of Rainfall Intensity-duration-frequency (IDF) Information: Guideline for Canadian Water Resources Practitioners, 2010.

Dawson, C. W., Abrahart, R. J. and See, L. M.: HydroTest: A web-based toolbox of evaluation metrics for the standardised assessment of hydrological forecasts, Environ. Model. Softw., 22(7), 1034–1052, doi:10.1016/j.envsoft.2006.06.008, 2007.

Deepthi Rajsekhar, Vijay P. Singh and Ashok K. Mishra: Hydrologic Drought Atlas for Texas, J. Hydrol. Eng., 20(7), doi:10.1061/(ASCE)HE.1943-5584.0001074, 2015.

Deng, Z., Qiu, X., Liu, J., Madras, N., Wang, X. and Zhu, H.: Trend in frequency of extreme precipitation events over Ontario from ensembles of multiple GCMs, Clim. Dyn., 46(9–10), 2909–2921, 2016.

Fischer, E. M. and Knutti, R.: Observed heavy precipitation increase confirms theory and early models, Nat. Clim. Change, 6(11), 986–991, 2016.

Ganguli, P. and Reddy, M. J.: Probabilistic assessment of flood risks using trivariate copulas, Theor. Appl. Climatol., 111(1–2), 341–360, doi:10.1007/s00704-012-0664-4, 2012.

Gelman, A., Shirley, K. and others: Inference from simulations and monitoring convergence, Handb. Markov Chain Monte Carlo, 163–174, 2011.

Gu, X., Zhang, Q., Singh, V. P., Xiao, M. and Cheng, J.: Nonstationarity-based evaluation of flood risk in the Pearl River basin: changing patterns, causes and implications, Hydrol. Sci. J., 62(2), 246–258, 2017.

Hamed, K. H. and Rao, A. R.: A modified Mann-Kendall trend test for autocorrelated data, J. Hydrol., 204(1), 182–196, 1998.

Hu, S.: Akaike information criterion, Cent. Res. Sci. Comput., North Carolina State University. Available from: http://www4.ncsu.edu/~shu3/Presentation/AIC_2012.pdf, 2007.

Janga Reddy, M. and Ganguli, P.: Application of copulas for derivation of drought severity–duration–frequency curves, Hydrol. Process., 26(11), 1672–1685, doi:10.1002/hyp.8287, 2012.

Karmakar, S. and Simonovic, S. p.: Bivariate flood frequency analysis. Part 2: a copula-based approach with mixed marginal distributions, J. Flood Risk Manag., 2(1), 32–44, doi:10.1111/j.1753-318X.2009.01020.x, 2009.

Karmakar, S. and Simonovic, S.: Flood Frequency Analysis Using Copula with Mixed Marginal Distributions, Water Resour. Res. Rep. Available from: http://ir.lib.uwo.ca/wrrr/19, 2007.

Katz, R. W. and Brown, B. G.: Extreme events in a changing climate: variability is more important than averages, Clim. Change, 21(3), 289–302, 1992.

Katz, R. W., Parlange, M. B. and Naveau, P.: Statistics of extremes in hydrology, Adv. Water Resour., 25(8), 1287–1304, 2002.

Komi, K., Amisigo, B. A., Diekkrüger, B. and Hountondji, F. C.: Regional Flood Frequency Analysis in the Volta River Basin, West Africa, Hydrology, 3(1), 5, 2016.

Mailhot, A., Duchesne, S., Caya, D. and Talbot, G.: Assessment of future change in intensity–duration–frequency (IDF) curves for Southern Quebec using the Canadian Regional Climate Model (CRCM), J. Hydrol., 347(1), 197–210, 2007.

Prein, A. F., Rasmussen, R. M., Ikeda, K., Liu, C., Clark, M. P. and Holland, G. J.: The future intensification of hourly precipitation extremes, Nat. Clim. Change, advance online publication, doi:10.1038/nclimate3168, 2016.

Sadri, S., Kam, J. and Sheffield, J.: Nonstationarity of low flows and their timing in the eastern United States, Hydrol Earth Syst Sci, 20(2), 633–649, 2016.

Serinaldi, F. and Kilsby, C. G.: Stationarity is undead: Uncertainty dominates the distribution of extremes, Adv. Water Resour., 77, 17–36, 2015.

Ter Braak, C. J. and Vrugt, J. A.: Differential evolution Markov chain with snooker updater and fewer chains, Stat. Comput., 18(4), 435–446, 2008.

Ter Braak, C. J.: A Markov Chain Monte Carlo version of the genetic algorithm Differential Evolution: easy Bayesian computing for real parameter spaces, Stat. Comput., 16(3), 239–249, 2006.

Wang, X., Huang, G., Liu, J., Li, Z. and Zhao, S.: Ensemble projections of regional climatic changes over Ontario, Canada, J. Clim., 28(18), 7327–7346, 2015.

Wang, X., Huang, G., Liu, J., Li, Z. and Zhao, S.: Ensemble projections of regional climatic changes over Ontario, Canada, J. Clim., 28(18), 7327–7346, 2015.

Xie, H., Li, D. and Xiong, L.: Exploring the ability of the Pettitt method for detecting change point by Monte Carlo simulation, Stoch. Environ. Res. Risk Assess., 28(7), 1643–1655, 2014.

Yilmaz, A. G., Hossain, I. and Perera, B. J. C.: Effect of climate change and variability on extreme rainfall intensity–frequency–duration relationships: a case study of Melbourne, Hydrol. Earth Syst. Sci., 18(10), 4065–4076, 2014.

Yilmaz, A. G., Imteaz, M. A. and Perera, B. J. C.: Investigation of non-stationarity of extreme rainfalls and spatial variability of rainfall intensity–frequency–duration relationships: a case study of Victoria, Australia, Int. J. Climatol., 37(1), 430–442, doi:10.1002/joc.4716, 2017.

Yue, S. and Wang, C. Y.: Power of the Mann–Whitney test for detecting a shift in median or mean of hydro-meteorological data, Stoch. Environ. Res. Risk Assess., 16(4), 307–323, 2002.

---

## Author Comment (AC2) · 16 Aug 2017

**Responses to Reviewer #2 on "Does Nonstationarity in Rainfall Requires Nonstationary Intensity-Duration-Frequency Curves? By Poulomi Ganguli and Paulin Coulibaly**

We thank Referee #2 for reviewing our manuscript and providing constructive feedback. Our responses are embedded within the comments (in BLACK) in BLUE.

**Reviewer #2**

**Comment 1.** The manuscript could do with a good proof read and rewrite. There are lots of little mistakes which makes the paper very difficult to read. I was constantly stopped in my train of reading by small errors or references to figures/tables which weren't explained. The supplementary material is 66 pages and has 37 Tables. This is huge and difficult to come to terms with – I couldn't follow it all. As I don't believe a specific structure is required can I recommend the following? Group the supplementary text and figures and tables into sections. That way you will have separate sections to refer to in the main text. You can then go sequentially through the text. S1 is the infilling, S2 is the autocorrelation method and results, S3 non-stationarity test method and results, S4 GEV fitting. I may have got the headings incorrect but I hope what I mean is clear. Then with the results you can just reference a section for detailed results and focus on discussing the figures in the main text. Trying to interpret 37 tables (some split into two) – almost all which are referenced in the main text - it is like trying to read a thesis.

**Response:** This is indeed a good point and we have revised the supplementary section and reorganized the material into various sections as suggested. We discussed corresponding results in the form of tables and figures under each subsection making it more coherent and easier to read. Also, we have moved some of the Tables (for example, Table S1) from supplements to main manuscript reducing the length of the Supplements to 57 pages with 30 tables all together.

**Comment 2.** Moving Table S1 to the main text, and maybe removing Figure S1 altogether will make the manuscript more standalone and easier to read. This manuscript is a bit short on doing justice to some of the previous work done in this area.

**Response:** Here we partially agree with the reviewer's comment. We have moved Table S1 to the main text. However, we have retained the Figure S1 in the Supplement since the figure provides a conceptual representation of changes in probability density functions of extremes in a nonstationary environment. We feel the figure will help readers in understanding how the nonstationarity may lead to changes in the distribution of extremes, which can potentially lead to the changes in the frequency of extremes.

**Comment 3.** Page 2 – Line 21: This is the only line discussing previous work to do with non-stationary IDFs. I think this work deserves more attention given that the focus of this manuscript is non-stationary IDFs. My recommendation is as follows:

In Page 1 – Line 23: "In a warming climate . . ." I would be a bit more careful here and expand this. I would cite Lenderink and van Meijgaard (2008) and Wasko and Sharma (2015) as papers that link temperature increases to intensifying rainfall. Most of the papers cited at the end of this sentence deal with temporal precipitation trends (and not necessarily links to temperature). It is important to make that distinction.

The reason I make the above point is the covariate used for non-stationarity is important. The authors don't raise this till the second last of their manuscript citing Mondal and Mujumdar (2015). This needs to come up in the introduction to put this manuscript novelty in context. There are more papers in this space. For example Agilan and Umamahesh (2017) and Ali and Mishra (2017) who argue for temperature to be used as a covariate (and not necessarily time). Indeed Wasko and Sharma (2017) show that temperature is a good covariate when predicting future rainfall. Other work by Agilan and Umamahesh may also be relevant and should be discussed. Finally, I am pretty sure at least one of the Yilmaz papers suggests not much evidence (if any) for using non-stationarity so in the introduction this is not cited correctly (though I note in the discussion it is). To summarise – the literature review needs to be expanded on the above point.

**Response:** Agreed. We expanded the literature review section in the revision. We add following sentences in the revision:

"For sub-hourly and up to six-hourly extreme precipitation, increases at or above the C-C rate have been found in the Netherlands (Lenderink and van Meijgaard, 2008; Lenderink et al., 2017), Switzerland (Ban et al., 2014), Germany (Berg et al., 2013), the UK (Blenkinsop et al., 2015), the Mediterranean (Drobinski et al., 2016), most of Australia (Wasko and Sharma, 2015, 2017), North America (Shaw et al., 2011) and China (Miao et al., 2016), while in India (Ali and Mishra, 2017) and northern Australia (Hardwick Jones et al., 2010) negative rates have been observed. The extent of urbanization also contributes to extreme regional precipitation through urban heat island effect and aerosol concentration (Dixon and Mote, 2003; Mölders and Olson, 2004; Nihongi et al. 2007; Mohsen and Gough, 2012; Wang et al., 2015). Agilan and Umamahesh (2017) incorporated six physical processes, namely, time, urbanization, local temperature changes, annual global temperature anomaly (as an indicator of global warming), El Niño-Southern Oscillation (ENSO) and Indian Ocean Dipole (IOD) as covariates in the nonstationary GEV models for analyzing extreme precipitation in the city of Hyderabad, India. Their analysis indicated that the local processes, urbanization and local temperature changes are the best covariates for short-duration rainfall, whereas global processes, such as, global warming, ENSO cycle and IOD are the best covariates for the long duration rainfall. In their study, however, time was never qualified as the best covariate for modeling local scale extreme rainfall intensity. Singh et al. (2016) performed nonstationary frequency analysis of Indian Summer Monsoon Rainfall extreme (ISMR; defined as cumulative rainfall over continental India during 1 June to 30 September) and found evidence of significant nonstationarity in ISMR extremes in urbanizing/developing-urban areas (transitioning from rural to urban), as compared to completely urbanized or rural areas.

However, their analysis was performed at a spatial resolution of 1° using gridded daily precipitation data obtained from Indian Meteorological Department (IMD). Ali and Mishra (2017) showed that a strong (higher than C-C rate) positive relationship exists between 3-hourly and daily rainfall extremes and dew point, and tropospheric temperature (T850; or the temperature in the upper troposphere at 850 hPa) over 23 urban locations in India. The latter two were subsequently used as covariates for nonstationary design storm estimates. The results indicated an increase in rainfall maxima at a majority of locations assuming nonstationary conditions over stationary atmospheric conditions. In contrast, in another studies, over Melbourne and Victoria, in Australia, Yilmaz et al. (2014; 2017) found superiority of stationary models over nonstationary models. Yilmaz et al. (2014; 2017), considered both nonstationarity in time and large scale climate oscillations affecting Australian rainfall in their analyses. However, most of these previous studies have analyzed changes in expected point estimates of nonstationary versus stationary Design Storm Intensity (hereafter referred as DSI), but have not reported the statistical significance of the difference between two methods of estimates". To our best knowledge, no thorough comparison of stationary vs. nonstationary methods for deriving IDF statistics has been conducted in Southern Ontario.

**Comment 4.** Another problem I have is with the paragraph on Page 3 that starts with "secondly" – I don't think any of the research questions actually address the "secondly" point. Reading page 7 it seems you adopt the GEV and don't necessarily test this is a better fit than other distribution. This is fine – but the way this paragraph sets up the reader for something else. Either omit the "secondly" paragraph altogether or add another point to the bottom of Page 3 saying you use a GEV and the reason for doing so.

**Response:** This is indeed a good point. Agreed! We have re-organized this section and moved limitations of GEV in subsection 3.3 (lines 15 – 22) in section 3. The choice of the GEV was based on a previous study where various distribution functions were compared in the study area (Switzman et al. 2017).

**Comment 5.** You introduce the EC data without context – so I had no idea why it was there until I got to page 11.

**Response:** We appreciate the reviewer's point. We have introduced few sentences in the introduction section (page 4, line 20-23) to highlight the rationale behind the inclusion of EC data. We argue that:

"… so far very few studies have reported the difference between the updated versus EC generated IDF, taking into account nonstationarity in design consideration. Simonovic and Peck (2009) compared updated versus EC IDF for the city of London, Ontario and reported EC IDF curves shows a difference of the order of around 20%. However, their analysis was based on the stationarity assumption of precipitation extremes." Similarly, Coulibaly et al.

2015 have compared EC-IDF with stationary GEV based IDF across southern Ontario, no nonstationary methods were investigated.

**Comment 6.** Top of Page 11 reads like a discussion and seems squished between the presentation of results in Figure 5 and 6. You could consider a separate discussion section and reordering of the text.

**Response:** Agreed. We have moved this part of the text to Discussion and Conclusion section.

**Other comments:**

Page 2 – Line 16: If you are to introduce an abbreviation (TBRG) it helps to capitalise the first letter in each word before the abbreviation. This happens at several points in the text – I won't comment on the other occurrences.

**Response:** Agreed. We have capitalized the first letter in each word before the abbreviation for TBRG and other words in the revised version of the manuscript.

Page 2 – Line 22: "The nonstationary behaviour. . ." I think I would expand this sentence to just state what places/regions the citations have studied. Reason being – in the abstract and following sentences you are referring only to Canada – so when I get to this point I am not sure if you are being Canada specific or not. Maybe this should be the start of a new paragraph and expanded a bit.

**Response:** Agreed. As suggested we have expanded this sentence to include list of regions where the citations have studied in Page 4, lines 6 – 13. We also started this in a new paragraph as suggested. We have added following sentences in the revision:

"The nonstationary behavior of rainfall extremes is already being reflected in the increase in frequency or magnitude of such events, resulting in a shift of its distribution [Figure SPM 0.3 in Intergovernmental Panel on Climate Change Special Report on Extremes, IPCC SREX Report: Field, 2012; Fig S1: IPCC AR5 working Group Report, (Stocker et al., 2013)]. For instance, seasonal and annual extreme precipitation events in north-central and eastern US in 2013 (Knutson et al., 2014); extreme rainfall events in the Golden Bay region in New Zeeland (Dean et al., 2013); increase in precipitation rate in northern Europe (Yiou and Cattiaux, 2013), successive winter storm events in southern England in 2013/2014 leading to severe winter floods (Schaller et al., 2016), are primarily attributable to intrinsic natural variability and partly to anthropogenic influences."

Also, in Page 2, lines 1-6, we list the places where increase/decrease in extreme precipitation is linked to C-C scaling. We have added following sentences:

"For sub-hourly and up to six-hourly extreme precipitation, increases at or above the C-C rate have been found in the Netherlands (Lenderink and van Meijgaard, 2008; Lenderink et al.,

2017), Switzerland (Ban et al., 2014), Germany (Berg et al., 2013), the UK (Blenkinsop et al., 2015), the Mediterranean (Drobinski et al., 2016), most of Australia (Wasko and Sharma, 2015, 2017), North America (Shaw et al., 2011) and China (Miao et al., 2016), while in India (Ali and Mishra, 2017) and northern Australia (Hardwick Jones et al., 2010) negative rates have been observed."

Page 2 – Line 26: What result? This sentence doesn't make sense – maybe some expansion of the sentences here would help.

**Response:** Agreed. We have revised this sentence as:

"The asymmetric changes in the distribution of extremes owing to climate change have been subsequently validated for winter temperature extremes over the northern hemisphere (Kodra and Ganguly, 2014), and regional short duration precipitation extremes in India and Australia (Mondal and Mujumdar, 2015; Westra and Sisson, 2011)".

Page 3 – Line 7: Replace "secondly" with "The second drawback of IDF curves is". You have written too much to have just the word "secondly" here. Stylistically, I don't think "first", "second" etc need to be in italics. Particularly at the bottom at Page 3 – if you are that keen on this maybe a bullet point list would be better?

**Response:** We have revised this section in the current version of the manuscript.

Page 4 - Line 1: Remove "secondly".
**Response:** Agreed and incorporated as suggested.

Page 5 – Line 6: The reference to Table S1 doesn't belong here. I also believe Table S1 belongs in the main text.

**Response:** Agreed. Table S1 is moved to the main manuscript.

Figure 1 – Are the record lengths for daily or sub-daily? I don't think the caption says which.

**Response:** Agreed and we have revised the caption accordingly. This includes hourly, sub-hourly and daily record, which we together termed as short-duration Annual Maxima Precipitation (AMP) record.

Page 5 – Line 26 – "Imputation" isn't the correct word I don't think. Infilling maybe?

**Response:** Agreed and incorporated as suggested.

Page 6 – Line 21 – Stylistically, why don't you just say "Tables S2-S4"? I do feel if you composed the supplementary material in sections you could say section S1 and be done with it.

**Response:** Agreed and incorporated.

Page 6 – Line 24 – "Figure 2 shows . . ." You are repeating a previous a sentence Section 3.2 – Is the KPSS test in Figure 2?

**Response:** Agreed. We have included KPSS test in the flowchart.

Page 8 – Line 4 – who else makes this assumption that only the location and scale parameter vary? I know other authors make this assumption so this assumption needs to be put in context of the other work done in this area.

**Response:** Agreed. We have included list of references that assumes location and scale parameter(s) vary. We have added the following sentences in page 10, line 21 in the revised manuscript:

"For nonstationary model, the shape parameter is assumed as constant throughout. Here it should be noted that for modeling temporal changes in $\zeta$ requires long-term observations, which are often not available in practice (Cheng et al., 2014). Hence, following previous studies (Cannon, 2010; Cheng et al., 2014; El Adlouni et al., 2007; Gu et al., 2017) we incorporated time-varying covariates into GEV location (GEV$_t$-I), and both in location and scale parameters (GEV$_t$-II) respectively, to describe trends as a function of time".

Page 8 – Line 18 – So I went to the supplementary material as the text recommends and I saw four models fitted for each duration but I wasn't sure which model was which. Could this section in the main text be rewritten (maybe use some sort of list?) to say what models were fitted and clearly state their abbreviation

**Response:** This section has been revised. Further, as suggested the abbreviations of models are included in page 10, line 22 and in the footnote of Table 4.

Page 8 – Line 26: I disagree. Skewness of a distribution does not indicate a temporal trend. This a good example of a vague sentence with a Figure in brackets (in this case Figure 3) but no mention of what I am meant to get out of looking Figure 3 in reference to this sentence. This happens throughout the text.

**Response:** Agreed. We have revised the sentence as follows:

"The skewness is a measure of the asymmetry in the AMP distribution. Positive values of skewness indicate that data are skewed to the right."

Further, we have removed such inconsistencies in the revised version of the manuscript.

Figure 3 – your caption says hourly and sub-hourly. The headings in the captions go up to daily. You say you did statistical tests at 5 and 10% but don't say which final significance is presented in the plot. A legend wouldn't go astray . . .

**Response:** We have revised the Figure 3 caption as suggested.

Figure 4 – is there a particular time used for the non-stationary plots?

**Response:** This comment was not clear to us. Nevertheless, we have revised the Figure 4 caption to avoid any ambiguity. We have revised our figure caption as:

"DSI estimates of median (horizontal line within the box plot) and 95% credible intervals for 100-year return periods of stationary versus nonstationary models (a - i). The boxplots indicate the uncertainty in estimated DSI using Bayesian inference."

Page 10 – Maybe I missed this somewhere but what is the "z-statistic"? Is this the statistical test for the difference between two means?

**Response:** The reviewer is correct. The z-statistic is the test score for the difference between two means. We clarify this procedure in the Supplementary section of the revised manuscript.

Figure 6 – Should this have a negative scale too? Are there some sites which decrease?

**Response:** The reviewer is correct. We have added color map for negative scale too in the revised manuscript.
* * *

[revised manuscript text omitted]

---

## Author Response (AR3)

**Responses to Reviewers on "Does Nonstationarity in Rainfall Requires Nonstationary Intensity-Duration-Frequency Curves? By Poulomi Ganguli and Paulin Coulibaly**

We thank the Referees for reviewing our manuscript and providing constructive feedback. Our responses are embedded within the comments (in BLACK) in BLUE. The new additions to the revised manuscript are embedded below in GREEN.

**Report #1**

**Comment 1:** These papers are relevant to the discussion on nonstationarity in flow responses as it appears to be getting caused by antecedent conditions. Please include them in your discussion.

Ivancic, T. J., and S. B. Shaw (2015), Examining why trends in very heavy precipitation should not be mistaken for trends in very high river discharge, Clim. Change, 133(4), 681–693, doi:10.1007/s10584-015-1476-1.
Wasko, C., and A. Sharma (2017), Global assessment of flood and storm extremes with increased temperatures, Sci. Rep., 7(1), 7945, doi:10.1038/s41598-017-08481-1.

**Response:** Agreed. We have incorporated following sentences in the discussion (Page 19, lines 6 – 20):

".. review of the literature suggests that heavy precipitation does not necessarily lead to high stream discharge (Ivancic and Shaw, 2015; Do et al., 2017; Wasko and Sharma, 2017). The analysis of Do et al. (2017) reveals the trend in streamflow is more consistent across continental scale and neither the anthropogenic activities such as the presence of dams nor the vegetation cover had any significant effect on the trend estimates. Interestingly, the consensus among all three studies is that the catchment size, which regulates the flow response because of antecedent moisture content, is the most important contributing factor modulating the nature of trend in stream discharge. The smaller (especially, urban) catchments may have increased flood peaks, in contrast, larger (agricultural and rural) catchments may experience decreased runoff due to lower soil moisture. This can be attributed to the fact that high temperature leads to drying up of soil more quickly in larger catchments, thus forcing a large portion of precipitation not to become an overland flow. Finally, using more than 9000 daily streamflow records globally, Do et al. (2017) showed more stations with significant decreasing trends in annual maximum streamflow than that of significant increasing trends, indicating limited evidence of increasing flood hazards. Their findings corroborated with (Wasko and Sharma, 2017), in which authors showed that only in most extreme cases, for small catchments, increase in precipitation at higher temperature leads to increase in streamflow."

**Minor comments:**

**Comment 2:** This manuscript quite nicely shows that detectable non-stationarity does not necessarily lead to significant differences in design storm values. A very valuable contribution.

Figure 5 is excellent in conveying this message. The structure is much improved and the manuscript flows well but still has a number of instances of incorrect grammar.

**Response:** We appreciate the reviewer's positive feedback. We have added the first sentence of the comment in the discussion and conclusion section. We did one more round of proof reading.

**Comment 3:** Page 2 – Line 1 and Line 6 – the Wasko and Sharma (2017) paper cited here looks at using temperature as a covariate for non-stationarity and should be cited further down on this page (see original review comments).

**Response:** We have added following sentences in the revision (Page 3, lines 2 – 4):

"Using temperature as a covariate for nonstationarity, (Wasko and Sharma, 2017) investigated the sensitivity of extreme daily precipitation and streamflow to changes in daily temperature. Their results suggested little evidence of increase in stream flow with increases in heavy rainfall events at higher temperature".

**Comment 4:** Sorry that I missed this before but I am surprised that Cheng and AghaKouchak (2014) is not introduced in the introduction given I think it is the first manuscript looking at non-stationary IDFs. It probably should be the first paper introduced in terms of this part of the introduction.

**Response:** Agreed and incorporated in the introduction (Page 2, lines 10-11) as below:

"One of the first attempt to derive nonstationary IDF through Bayesian Inference approach for extreme value analysis was by Cheng and AghaKouchak (2014), where authors introduced a linear trend in the parameters of the selected distribution."

**Comment 5:** Figure 1b – I don't really understand the population size scale – is it meant to be density?

**Response:** We concur with the reviewer. The population size indicates – 'density' and incorporated in the revision.

**Comment 6:** Page 6 - Line 1: This is repeating word for word the legend of the figure. Doesn't need to be here.

**Response:** Agreed and repetition was removed.

**Comment 7:** Page 9 – Line 27, Page 12 – Line 12: Maybe I got this wrong, but aren't these sentence contradictory? One suggests left skew, the other right skew.

**Response:** We agree and revised the sentence in Page 10 – line 9 as below:

"However, the short-duration AMP intensities often exhibit fat-tailed behavior, indicating large skewness and kurtosis."

**Comment 8:** Upon reading this paper again I would remove Table 2 and Table 3 – or move to supplementary. I don't think they add much to the manuscript and the discussion of them is only one or two sentences.

**Response:** Agreed. Keeping in view of both reviewers, we have removed Table 2 and 3 from the revised version.

**Comment 9** Table 4-7: Need to define LB and UB, need to put in the table caption why some of the values are bold.

**Response:** We thank the reviewer for raising this issue. However, please note that all explanations including definitions of LB and UB are provided as a footnote of Tables 2 - 5 (in main manuscript) and Table S13 (in the Supplement).

**Comment 10:** Page 14 – Line 21: More or less subtle than those for the 100 year.

**Response:** Agreed and incorporated in the manuscript.

**Comment 11** Page 16 – Line 1: Figure or Figures?

**Response:** Agreed and incorporated in the manuscript.
* * *
**Report #2**

**Comment 1:** I appreciate the authors' effort to improve the presentation of the material, but the paper to me still feels like an overwhelming amount of test results which are only partially digested and made sense of. The authors obviously disagree with my opinion - we'll have to agree to disagree on this one. I still think that too much information necessary to understand the paper is given in the supplementary material and is difficult to identify due to the enormous amount of information the authors share with the reader.

**Response:** Thanks for the constructive comments. We gently disagree with the reviewer's point. As explained in our earlier responses to the reviewer comments, we re-iterate again that some of these tests are complimentary to each other. Secondly, in response to reviewer Comment 2, we have organized the supplementary material section wise for smooth understanding of the reader.

**Comment 2:** I think the authors have only partially replied to some of comments - although the new manuscript does address many of my concerns and improves the presentation. I would still recommend to the authors to have one more round of proof-reading: there are still many grammatical mistakes.

**Response:** We appreciate the reviewer's feedback. We tried our best to address many of the reviewer's comments in order to improve the manuscript from its previous version. We did one more round of proof-reading to further correct grammatical errors.

**Comment 3:** I will just here make some further comments, so are re-iteration of some of the comments which I feel the authors did not do full justice to. I believe these concerns relate to the very core of the analysis and question the validity and robustness of the findings in the manuscript.

When I said the authors do not fit IDF curves I meant that as far as I can tell there is not attempt made in the estimation procedure to ensure that design events for longer durations are larger or equal to design events of a shorter duration. I would expect the intensity of 100-year event for the 30-minute event to be <= than the 100-year event for the 15-minute event. In the plots shown by the authors (for example in Figure 4) this is the case mostly (but not for the Startford WWTP station when looking at the 15- and 30-minute boxplots), but the modelling does not enforce this. Enforcing the consistency of the IDF curves is one of the challenges of the joint modelling of rainfall of different durations, which the authors don't tackle in my opinion (and that's fine as long as this is acknowledged). This lack of shape enforcement is also the cause of the funny kinks seen in Figure 8 and S15, which are the result of fitting separate distributions to the series of different durations. The authors disagree with my take on the issue in Page 2 of the "Authors' replies" document, where they state they have developed IDF curves.

**Response:** We agree. We have added the following sentences in Page 20, lines 20-24, to indicate this as a limitation of our study:

"One of the limitations of the present analysis includes the lack of accounting for the consistency of the IDF curves in terms of shape enforcement. The lack of shape enforcement in the IDF curves (Figures 8 and S15) is the result of fitting separate distributions to the series of different durations."

**Comment 4:** In their reply to my comment given in Page 3 of the "Authors' replies" document I feel the authors have not really replied to my Comment: they just re-iterate the fact that they use a lot of tests as it was previously done in the literature, but do not acknowledge the possible issues connected to multiple comparisons. The moment in which one begins to do a lot of statistical tests is exactly the time where the issue of multiple comparisons becomes an issue. It is ok to do several tests on your data, but you are bound to have some false positives. Just adding more tests in the mix will not help necessarily in having a clearer view of possible changes in the series, as there will necessarily be some spurious result due to randomness.

**Response**: This is indeed a very good point raised by the reviewer. In order to verify the issue of multiple comparisons, here we discuss adjusted False Discovery Rate (FDR) based p-values for a representative station (*i.e.*, Hamilton; Table 1), and overall summary results for all durations ($d$ = 15-min, …., 24-hour) and station locations (Table 2).

**Table 1**. FDR-adjusted *p*-value for Hamilton across all durations. Highlighted cells indicate test statistics with adjusted p-value < 0.10

| Duration | Ljung-Box Test | KPSS Test | Mann-Kendall Trend Test | Priestley-Subbarao Test | Pettitt Test |
|---|---|---|---|---|---|
| 15-min | 0.43 | 0.43 | 0.52 | 0.52 | 0.52 |
| 30-min | 0.91 | 0.25 | 0.39 | 1.28E-04 | 0.50 |
| 1-hr | 0.92 | 0.25 | 0.92 | 0.01 | 0.92 |
| 2-hr | 0.74 | 0.11 | 0.38 | 4.02E-05 | 0.38 |
| 6-hr | 0.18 | 0.05 | 0.98 | 1.42E-05 | 0.28 |
| 12-hr | 0.25 | 0.06 | 0.90 | 1.39E-09 | 0.63 |
| 24-hr | 0.40 | 0.03 | 0.75 | 0.00489 | 0.40 |

**Table 2** Number of test statistics (out of five tests) in which FDR-adjusted p-value < 0.10

| duration | Toronto | Hamilton | Oshawa | Windsor | Kingston | London | Trenton | Stratford | Fergus shand |
|---|---|---|---|---|---|---|---|---|---|
| 15-min | | | | | | | 1 | | |
| 30-min | | 1 | | | | 1 | 1 | 1 | 2 |
| 1-hr | | 1 | | | | | 1 | 1 | 1 |
| 2-hr | | 1 | 1 | | | | 3 | 1 | 1 |
| 6-hr | | 2 | | | 2 | | 1 | 1 | |
| 12-hr | 1 | 2 | 1 | | 1 | | 3 | 1 | |
| 24-hr | | 2 | 3 | | 1 | 3 | 3 | 1 | 1 |

However, keeping the view of the reviewer's concern, in the main manuscript (or in the Supplementary), instead of adding more results, we have added following the lines in the Methodology (page 9; line 25) and Results (page 14; lines 6-9) sections:

"Although we present a range of statistical tests to investigate plausible shifts in the time series, we do acknowledge that statistical inferences may be affected by the problem of multiple comparisons resulting into a set of false positive outcomes. In order to test the issue of multiple comparisons, we analyzed p-values of five statistical tests, i.e., Ljung-Box test, KPSS test, Mann-Kendall trend test, Priestley-Subbarao test, and Pettitt test using False Discovery Rate (FDR) method (not shown here) as suggested by (Benjamini and Hochberg, 1995). However, we excluded ADF, Mann-Whitney and Mood tests from the analysis since unlike other tests, the higher p-value in ADF statistics indicate the presence of nonstationarity in the time series. On the other hand, the latter two tests do not offer any p-values".

"We find that except Windsor, in all sites the presence of trend and non-stationarities in the time series turns out to be significant with the highest number of statistical tests showing adjusted p-value < 0.10 in Trenton followed by Hamilton. This indicates our analysis is not affected by the issue of multiple comparisons."

**Comment 5:** In their reply to my further comment 7 given in Page 8 of the "Authors' replies" document I feel the authors have not addressed my point, but have just given some good reasons of why to use non-parametric tests when detecting changes in some series. Again, I am asking what is the point of doing all of these non-parametric tests if the information derived from the non-parametric tests is not used in any way the parametric modelling, which is the basis for all the comparison against the present day approach. If the Pettitt's test identify a change point should 't we also test the presence of step-changes in the parametric modeling. If we don't use the non-parametric tests in some way: why bother to perform them. My perception is that they conclusions mostly rely on the non-stationary GEV-fits with linear trends and the comparisons of the non-stationary DSI and EC curves. So again, why use a Pettitt test if they claim themselves in the reply to my comment "In practice, when real change point is unknown, often Mann-Whitney test, in general, does not work well and the Pettitt method can yield plausible change point location along with its statistical significance. However, the significance of the Pettitt test can be obtained using an approximated limiting distribution. Therefore, above tests were needed in the current setting." Why doing a Pettitt test and then not try to use a parametric model to express the potential step changes in the series?

**Response**: This is indeed a good point raised by the reviewer. In the present study, non-stationary is introduced in the GEV model through a trend on the location and scale parameter as a linear function of time. We find this is widely used method to model non-stationarity in the literature considering time as a covariate via a linear or a polynomial function (Singh et al., 2016; Villarini et al., 2009). An alternative representation of nonstationarity can be obtained by assuming change point in the GEV parameters (Renard et al., 2013). However, given the scope of the present work, we wish to expand our analysis considering step changes in a parametric GEV models in near future.
Therefore, we have added the following sentences as part of the limitations of the present work in page 20, lines 23-24 and 28:

"Although in several instances, we find evidence of step changes in short-duration rainfall extremes, we have not introduced any change point model in the GEV parameters (Renard et al., 2013). Future research is directed towards including the step-change model in GEV location and scale parameters."

**Comment 6:** Overall I feel the authors do a lot of analysis which they end up not using in the final modelling: for example they have included the Bayes factor but only rely on it when it gives evidence in favour of non--stationary models, using the AICc when BF doesn't do what they want it to do. It's redundant to the reader to give results which end up not being used to draw conclusions.

**Response**: We thank the reviewer for pointing out this. We have added following sentences in page 14, line 22 in the revised manuscript:

"Overall, results of Bayes factor values indicate that there is no strong evidence to favor or reject any of the three models. In general, although we find the stationary model cannot be rejected, it does not imply that there is no change. We may be unable to detect the apparent signal of nonstationarity due to the strong natural variability present in the data."

**Comment 7:** Regarding the AICc - the authors now make it clearer that they use a very specific form of the AIC, which is not the standard one: I welcome this further clarification, but would drop the reference to the Akaike 1974 paper which introduces the traditional AIC. It is not clear to me why the authors decide to use this version of the AIC rather than the traditional one - they don't really give a reason for their choice. I would also imagine that if they had used the standard version of the AIC maybe the outcome of using AIC and BF would have been slightly more in agreement, since they both use some form of the likelihood.

**Response**: We agree and drop the citation of Akaike 1974 paper. We also point to the reviewer that for small sample sizes (*i.e.*, $n/m < \sim 40$ in Eqn. 3.9), the second-order Akaike Information Criterion (AIC$_c$) should be used. As sample size increases, the last term of the AIC$_c$ approaches zero, and the AIC$_c$ tends to give the same conclusion as the traditional AIC (Burnham and Anderson, 2003). Since in the present study, length of data series varies from 46 to 66 years, we used AIC$_c$ instead of traditional AIC, which is also a widely used method for model selection in hydrology (Caroni and Panagoulia, 2016; Gu et al., 2017; Panagoulia et al., 2014). We have added these details in the revised supplementary document (in SI 3.2.2).

**Comment 8:** Comment 11 (Page 9 of the "Authors' replies" document): I thanks the author for the explanation, although if understand correctly, the 100-year event as it calculate now would be the biggest event in the 100 year starting from the year in which recording began. So say \mu_1 is positive and \sigma is constant, we have the 100-year event to coincide with the median of the posterior distribution of the 100-year event for the time t_100. This implies that the authors are extrapolating the effects detected with the current series to the future, and also does not somehow take into account that we would need some decades before hitting that maximum value for the 100-year event. There is quite a bit of research on how to update design event concept for changing extremes (for example Rootzen and Katz), which is a further challenge for engineers after trends have been detected in hydrometric series.

**Response**: We appreciate author's comment. We discuss the design event concept in the discussion part of the manuscript (in page 19; lines 22 – page 20, line 8) as below:

"The statistical uncertainty in modeling nonstationarity can result from multiple sources. For instance, extrapolating the effects detected with observed historical series to the more extreme values that have not yet been experienced, model choices resulting from selection of covariates in the nonstationary distributions (Agilan and Umamahesh, 2017), and the treatment of nonstationarity introduced through either a linear (Ali and Mishra, 2017; Cheng et al., 2014;

Westra et al., 2012) or polynomial (Villarini et al., 2009) trend, or a change-point (Renard et al., 2013) in the model. The key question remains how to update design events in a nonstationary climate. This becomes further challenging after trends and change points are detected in hydrometric time series. To address climate change adaptation need under nonstationarity and uncertainty, some of the concepts discussed in the literature are design life level (Rootzén and Katz, 2013) to quantify the probability of exceeding a fixed threshold during the design life of a project, replacing the commonly used concept of average return period with reliability (Read and Vogel, 2015) and a risk-based decision-making approach, integrating the concept of *expected regret* (Rosner et al., 2014). However, apart from statistical uncertainty, one of the important sources of uncertainty in future planning period is the use of climate model output. Modeling nonstationarity in future time period is complicated by the choice of spatial resolution of climate models, lack of understanding of model physics due to different model choices and inherent uncertainties in climate model simulations resulting from different initial condition runs, which is especially apparent over regional scale and decadal planning horizon (Ganguli et al., 2017; Hawkins and Sutton, 2009; Kumar and Ganguly, 2017; Meehl et al., 2009)."

**Comment 9:** In Figure 3 it is still not very clear to me how the authors decide when to have a coloured triangle and when to include a cross. You perform several change-point tests, so does the triangle get a colour if any of the change-point tests is significant (and what it they give indication in of change points in different years?). Similarly, which non-stationaruty test warrants for a cross to appear in the triangle: this needs to be mentioned either in the caption or in the main text.

**Response**: Agreed. We have incorporated following sentences in the figure caption of the revised manuscript:

"A cross symbol in the triangle indicates nonstationarity detected through Priestley Subbarao test statistics."

**Comment 10** SI 2.2: After eq. 2.6: the term "power" has a specific meaning in statistics, which is actually almost the opposite of the p-value. Use the term "significance".

**Response**: Agreed and incorporated in the manuscript.

**Comment 11** SI 3.1: after eq. 3.3, Where should have a small w. Further (and more importantly) from the text it seems that authors are confused to what is a posterior and what is a likelihood in a Bayesian context: $p(\omega|y)$ and $p(\lambda|y,x)$ are posterior distributions, while $p(y|\omega)$ and $p(y|\lambda,x)$ are likelihood functions. In Bayesian inference one uses the

posterior to make inference (and not to infer parameters). The sentence as it stands now makes no sense.

**Response**: We apologize for the error. In Eqn. 3.3 $\boldsymbol{\omega}$ should be replaced by $\lambda$. We revised the sentences as below:

"The $p(\boldsymbol{\omega}|\boldsymbol{y})$ and $p\left(\lambda \mid \boldsymbol{y}, \boldsymbol{x}\right)$ in Eqns. 3.1 and 3.2 indicate resulting posterior distributions whereas $p(\boldsymbol{y}|\boldsymbol{\omega})$ and $p\left(\boldsymbol{y}_t \mid \lambda, \boldsymbol{x}(t)\right)$ denote likelihood functions."

**Comment 12:** I would move the sentence "(often referred to as the burn-in period)" to after the description of what the burn-in period is, i.e. the next sentence.
**Response**: Agreed and incorporated in the revision.

**Comment 13:** The authors never specify the priors used in the analysis as far as I can tell: this is a very relevant piece of information which is missing.

**Response**: Agreed and incorporated in the revision as below:

"The priors for the location and scale parameters are non-informative normal distributions, whereas prior for the shape parameter is a normal distribution with a standard deviation 0.3 (Renard et al., 2013) as used in the default option in NEVA package."

**Comment 14** SI 3.2.1: it is confusing to have the parameters being denoted with $\theta$, rather than $\omega$ and $\lambda$ as in the previous section.

**Response**: Agreed and revised as suggested.

**Comment 15** Typo in title Require not requires
**Response**: Agreed and incorporated in the revision.

**Comment 16** Line 1 of the abstract: no need for the word "increased", since "risen" is used.
**Response**: Agreed and incorporated in the revision.

**Comment 17** Page 12 Line 30: missing a (decrease) and drop the vice-versa. Alternatively, drop the (decrease) in line 29.
**Response**: Agreed and incorporated in the revision.

[revised manuscript text omitted]